# scDisInFact: disentangled learning for integration and prediction of multi-batch multi-condition single-cell RNA-sequencing data

Ziqi Zhang [1], Xinye Zhao[2,5], Mehak Bindra [3,5], Peng Qiu [4] & Xiuwei Zhang [1] ✉

Single-cell RNA-sequencing (scRNA-seq) has been widely used for disease studies, where sample batches are collected from donors under different conditions including demographic groups, disease stages, and drug treatments. It is worth noting that the differences among sample batches in such a study are a mixture of technical confounders caused by batch effect and biological variations caused by condition effect. However, current batch effect removal methods often eliminate both technical batch effect and meaningful condition effect, while perturbation prediction methods solely focus on condition effect, resulting in inaccurate gene expression predictions due to unaccounted batch effect. Here we introduce scDisInFact, a deep learning framework that models both batch effect and condition effect in scRNA-seq data. scDisInFact learns latent factors that disentangle condition effect from batch effect, enabling it to simultaneously perform three tasks: batch effect removal, condition-associated key gene detection, and perturbation prediction. We evaluate scDisInFact on both simulated and real datasets, and compare its performance with baseline methods for each task. Our results demonstrate that scDisInFact outperforms existing methods that focus on individual tasks, providing a more comprehensive and accurate approach for integrating and predicting multi-batch multi-condition single-cell RNA-sequencing data.

Single-cell RNA-sequencing (scRNA-seq) is able to measure the expression levels of genes in each cell of an experimental batch. This technology has been widely used in disease studies, where samples are collected from donors at different stages of the disease or with different drug treatments[1–5]. As a result, each sample's scRNA-seq count matrix is associated with one or more biological conditions of the donor, which can be age, gender, drug treatment, disease severity, etc. Meanwhile, datasets that study the same disease are often obtained in different batches, which introduce technical variations (also termed batch effects) across batches[6,7]. In practice, the available samples in the datasets of a disease study can originate from different conditions and batches, as arranged in Fig. 1a (right). We term such datasets as multi-

[1]School of Computational Science and Engineering, Georgia Institute of Technology, Atlanta, GA, USA. [2]School of Electrical and Computer Engineering, Georgia Institute of Technology, Atlanta, GA, USA. [3]School of Biological Science, Georgia Institute of Technology, Atlanta, GA, USA. [4]Department of Biomedical Engineering, Georgia Institute of Technology and Emory University, Atlanta, GA, USA. [5]These authors contributed equally: Xinye Zhao, Mehak Bindra. ✉e-mail: xiuwei.zhang@gatech.edu

**Fig. 1 | Overview of scDisInFact. a** scDisInFact is applied on multi-batch multi-condition datasets where count matrices from disease studies are obtained from different experimental batches and conditions. Human figure created with BioRender.com. **b** The neural network structure of scDisInFact. scDisInFact uses an encoding network (left) to learn the disentangled latent factors, and uses a decoding network (right) to generate gene expression data from the latent factors. It is designed for tasks including (1) batch effect removal (latent factors disentanglement), (2) condition-associated key genes detection, and (3) perturbation prediction. Neural network illustration adapted from LeNail[40].

batch multi-condition datasets. In such datasets, biological variations caused by condition effects exist between data matrices generated in the same batch but corresponding to different biological conditions, while technical variations caused by batch effects exist between data matrices from the same condition but different batches. Therefore, the differences among these data matrices are a mixture of batch effects (technical variation) and condition effects (biological variation), which complicates the process of fully utilizing the potential of these datasets.

In this paper, we consider a few computational challenges that need to be tackled when using multi-batch multi-condition datasets for disease study: (1) removing batch effect while preserving biological condition effect; (2) detecting key genes associated with biological conditions; (3) predicting unseen data matrices corresponding to certain conditions (the matrices with dashed borders in Fig. 1a (right)), also known as the task of perturbation prediction. Methods have been designed for each problem separately, but no existing method can solve the three problems jointly. In the following, we discuss existing methods for each problem and their limitations.

Most existing batch effect removal methods treat the differences between data from different batches solely as batch effects and remove them by aligning different batches into a common distribution in either the original gene expression space or the latent embedding space[8–12]. Applying these methods to data from multiple conditions can result in over-correction[13], where biological differences among batches are also removed along with batch effects. Recently, methods have been proposed considering the biological differences among batches. scINSIGHT[14] factorizes the scRNA-seq matrices into common and condition-specific modules using non-negative matrix factorization. However, the factorization framework is limited to one type of condition, while multiple types of conditions, such as age, gender, and drug treatment, can co-exist in the dataset[15,16]. In addition, scINSIGHT corrects the batch effect on the latent space and cannot predict gene expression matrices that are removed of the batch effect under various conditions. scMC[13] integrates data batches without removing the biological variations among batches, but it does not disentangle biological variations caused by condition effect from those that are shared

among batches, nor does it output key genes associated with the biological conditions.

Another problem in the field is predicting the scRNA-seq data under one condition using data from another condition, also known as the problem of perturbation prediction[17]. This is particularly useful when predicting disease progression or drug effects, under conditions where data are not collected. Existing methods for this task, such as scGen[17] and scPreGAN[18], do not account for batch effects between data matrices and assume differences in cell distribution between data matrices solely result from biological conditions, an assumption that does not hold for most real datasets. Furthermore, in practice, there is often more than one type of condition in the data, but existing methods are designed for only one type of condition. For example, conditions such as disease severity and treatment can exist at the same time in a disease study, but existing methods can not deal with both types of conditions at the same time.

Here, we propose scDisInFact (single cell disentangled Integration preserving condition-specific Factors), which is the first method that can perform all three tasks: batch effect removal, condition-associated key genes (CKGs) detection, and perturbation prediction on multi-batch multi-condition scRNA-seq dataset (Fig. 1a). scDisInFact is designed based on a disentangle variational autoencoder framework. It disentangles the variation within the multi-batch multi-condition dataset into latent factors encoding the biological information shared across all data matrices, condition-specific biological information, and technical batch effect. The disentangled latent space allows scDisInFact to perform two other tasks, the CKG detection and perturbation prediction, and to overcome the limitation of existing methods for each task. In particular, the disentangled factors allow scDisInFact to remove batch effect while keeping the condition effect in gene expression data. In addition, scDisInFact expands the versatility of existing perturbation prediction methods in that (1) it models the effect of multiple condition types and (2) it enables the prediction of data across any combination of conditions and batches within the dataset. We compared scDisInFact with scINSIGHT in terms of batch effect removal and CKG detection. As scINSIGHT does not perform perturbation prediction, we compared scDisInFact with scGen and scPreGAN in terms of perturbation prediction. We tested scDisInFact on simulated and real datasets[1–4], and found that it outperforms baseline methods across various tasks. Owing to its superior performance and multi-task capabilities, scDisInFact can be employed to comprehensively analyze multi-batch multi-condition scRNA-seq datasets, facilitating a deeper understanding of disease progression and patient responses to drug treatments.

## Results

### Overview of scDisInFact

scDisInFact is designed for a general multi-batch multi-condition scRNA-seq dataset scenario where samples are obtained from donors of different conditions and profiled in multiple experimental batches (Fig. 1a). We termed the category of conditions as the condition type, and the condition label of each condition type as a condition. For example, given a dataset that is obtained from donors with varying disease severity and genders, disease severity and gender are considered two separate condition types, while specific severity levels such as healthy control, mild symptoms, and severe symptoms are conditions. When there is more than one condition types, a condition can be the combination of labels, each label from a condition type. For each cell, scDisInFact takes as input not only its gene expression data, but also its batch ID and identified condition of each condition type.

scDisInFact is designed based on a variational autoencoder (VAE) framework[19,20] (Fig. 1b). In the model, the encoder networks encode the high dimensional gene expression data of each cell into a disentangled set of latent factors, and the decoder network reconstructs gene expression data from the latent factors (Fig. 1b). scDisInFact has

multiple encoder networks, where each encoder learns independent latent factors from the data. Using the information of biological condition and batch ID of each cell, scDisInFact effectively disentangles the gene expression data into the shared biological factors (shared-bio factors), unshared biological factors (unshared-bio factors), and technical batch effect.

The shared encoder learns shared-bio factors (green vector in Fig. 1b), which represent the biological variations within a data matrix that are irrelevant to condition effect or batch effects. Such variations usually are reflected as heterogeneous cell types in individual data matrices. An unshared encoder learns unshared-bio factors, which represent biological variations that are related to the condition effect. The number of unshared encoders matches the number of condition types, with each unshared encoder learning unshared-bio factors exclusively for its corresponding condition type (yellow vectors in Fig. 1b). For example, given a dataset of donors with different disease severities and genders, two unshared encoders are used, where one learns the disease severity condition and the other learns the gender condition. The technical batch effect is encoded as pre-defined one-hot batch factors (blue vector in Fig. 1b) which are transformed from the batch IDs of each cell. The batch factors are appended to the gene expression data before being fed into the shared encoder (Fig. 1b left) in order to differentiate the gene expression data of cells from different batches. The unshared encoders do not take as input the batch factors since its first layer is used for CKGs detection ("Methods"). The decoder takes as input the shared-bio factors, unshared-bio factors, and batch factors, and reconstructs the input gene expression data (Fig. 1b right). In order to guide the unshared encoders to extract biological variations that are only relevant to condition effects, a Gaussian mixture prior is used for the unshared-bio factors and a linear classifier network is attached to the output of each unshared encoder ("Methods"). The classifier is trained together with the unshared encoder to predict the conditions of the cells (Fig. 1b left).

With the disentangled latent factors, scDisInFact can perform the three tasks it promises (red text in pink boxes in Fig. 1b): (1) Batch effect removal. The learned shared-bio factors are removed of the batch effect and condition effect and can be used as cell embedding for clustering and cell type identification. Combining the shared-bio factors and any unshared-bio factors gives cell embeddings including the corresponding condition effects. (2) Detection of condition-associated key genes (CKGs) for each condition type. The first layer of each unshared encoder is designed to be a key feature (gene) selection layer of the corresponding condition type. The final weights of that layer can be transformed into the gene scores associated with the condition type ("Methods", Supplementary Fig. 1a). (3) Perturbation prediction. After scDisInFact is trained and the shared-bio factors, unshared-bio factors, and batch factors are learned, the perturbation prediction task takes the gene expression data of a set of cells under a certain condition and batch as input, and predicts the gene expression data of the same cells under a different condition in the same or different batch (Supplementary Fig. 1b). Supplementary Fig. 1b shows how to predict data in condition 2 and batch 1 given data in condition 1 and batch 6. We first calculate new unshared-bio factors and batch factors according to the predicted condition. The decoder then generates the predicted data matrix using the new unshared bio-factors and batch factors ("Methods"). It is worth noting that scDisInFact can perform perturbation prediction from any condition and batch to any other condition and batch in the dataset ("Methods").

The loss formulation, training procedure, and procedures to obtain the results for each task from a trained scDisInFact model are included in Methods.

### Testing scDisInFact on simulated datasets

We first validate scDisInFact on simulated datasets where we have ground truth information. We generated simulated datasets with 2

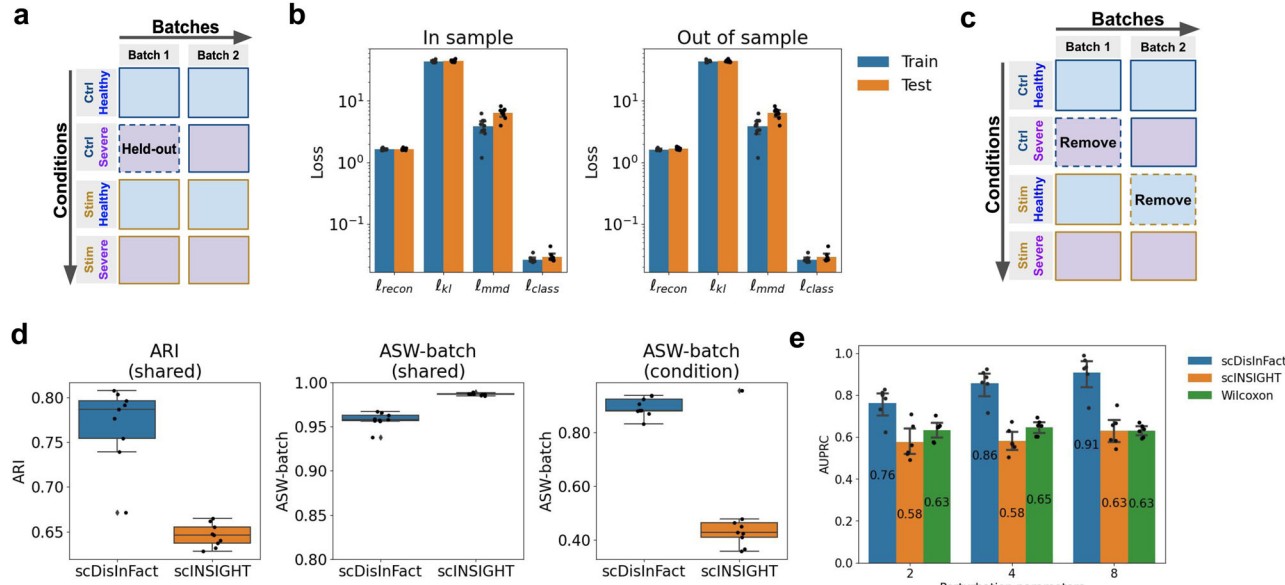

**Fig. 2 | Results on simulated datasets. a** The batches and conditions arrangement of count matrices in the simulated datasets. Totally 8 count matrices are generated in each dataset, corresponding to 2 batches and 2 condition types. The matrix with dashed borders is held out in the out-of-sample generalization test. **b** The loss values of the model in training and testing datasets, including both in-sample test (left) and out-of-sample test (right). $n = 9$ independent samples are included in each bar. The error bar represent 95% confidence interval, and the center of the error bar shows the mean. **c** The batches and conditions arrangement of count matrix in disentanglement test. The matrices with dashed borders are removed in the test. **d** The disentanglement scores of scDisInFact and scINSIGHT, including the ARI and ASW-batch scores for shared-bio factors and ASW-batch scores for unshared-bio factors. In the boxplots, the center lines show the median data value, and the box limits show the lower and upper quartiles (25% and 75%, respectively). The length of the whiskers is within 1.5x interquartile range. Outliers beyond the whiskers are plotted as points. $n = 9$ independent samples are included in each box. **e** AUPRC of scDisInFact, scINSIGHT, and Wilcoxon rank sum test on CKGs detection. $n = 6$ independent samples are included in each bar. The error bar represents 95% confidence interval, and the center of the error bar shows the mean. Source data for (**b**, **d**, and **e**) are provided in the Source Data file.

batches and 2 condition types including treatment and disease severity. The first condition type (treatment) includes cells under control (ctrl) and stimulation (stim) conditions, and the second condition type (severity) includes healthy and severe conditions. The condition and batch arrangement of count matrices in the dataset follows Fig. 2a. A total of nine datasets were generated with different numbers of CKGs and perturbation parameters, which models different strengths of condition effect ("Methods", parameter settings of each dataset follow Supplementary Table 1). In the following sections, we first test the generalization ability of scDisInFact, then show the performance of scDisInFact in terms of the three tasks, batch effect removal (which is also latent space disentanglement), key gene detection, and perturbation prediction.

One potential problem with neural network models is overfitting. If the model overfits, the model cannot be generalized and cannot be used for prediction tasks. Therefore, generalization ability is the basic requirement of scDisInFact in order to achieve successful disentanglement and make accurate perturbation predictions. We test the generalization ability of scDisInFact using the simulated dataset with 2 condition types (Fig. 2a).

We tested the generalization ability of scDisInFact under two different scenarios, according to the relationship between training and testing datasets, referred to as in-sample test and out-of-sample test. In the in-sample test, the training and testing data are evenly sampled from the same original dataset and they follow the same data distribution. We randomly held out 10% of cells in each count matrix as test data, and trained the model on the remaining 90% of cells. In the out-of-sample test, the training and testing data are not evenly sampled from the original dataset and the test distribution no longer matches the training distribution. We held out the count matrix corresponding to the condition *<ctrl, severe>* in batch 1, and trained the model using the remaining count matrices (Fig. 2a, held-out matrix

shown with a dashed border). We expect that it is harder for the model to generalize for the "out-of-sample" test due to the distribution difference.

After the model is trained on the training data, we tested it on the held-out data and compared the training and testing losses. The loss values on the 9 simulated datasets (Fig. 2b) show that in both scenarios (in-sample test and out-of-sample test), the testing losses are close to the training losses for various loss terms, confirming the generalization ability of scDisInFact.

We then tested the latent space disentanglement of scDisInFact on the simulated datasets, and compared its performance with scINSIGHT[14]. In practice, each batch often includes only a subset of the condition combinations. Therefore, to make the test case more realistic, we removed the count matrices corresponding to the *<ctrl, severe>* condition in batch 1 and the *<stim, healthy>* condition in batch 2 (Fig. 2c, removed matrices shown with dash borders). Since scINSIGHT can only work with one condition type, we created a "combined condition type" using the Cartesian product of the original condition types (including 4 combined conditions: "ctrl_healthy", "stim_healthy", "ctrl_severe", "stim_severe") and ran scINSIGHT using combined condition type. The calculation of shared-bio and unshared-bio factors for scINSIGHT is described in Methods.

A successful disentanglement requires that: (1) The shared-bio factors encode the cell-to-cell biological variations irrelevant to condition and batch effect. With these factors, we expect cells should be grouped according to cell types and aligned across batches and conditions. (2) The unshared-bio factors encode condition-specific biological variation across data matrices irrelevant to batch effect. With these factors, cells should be grouped according to conditions, and aligned across batches under each condition.

We evaluated the learned shared-bio factor in terms of batch effect removal (using ASW-batch score, "Methods") and separation of

cell types (using ARI score, "Methods"). The boxplots of these scores (Fig 2d, left and middle) show that scDisInFact has a significantly better performance in separating cell types while removing the batch effect compared to scINSIGHT. When evaluating the unshared-bio factors, we measured the grouping of conditions using ARI score, and the removal of batch effect using ASW-batch score ("Methods"). Both methods have perfect ARI scores (equal to 1), which is expected as both methods enforce the grouping in their objective function. In the meantime, scDisInFact achieves a higher ASW-batch score, and shows a better performance in removing the batch effect in the unshared-bio factors (Fig. 2d, right).

To visually assess the disentanglement results, we visualized the learned shared-bio factors and unshared-bio factors from one simulated dataset (Simulation 1 in Supplementary Table 1). We visualized the shared-bio factors using UMAP and observe that cell types are well-separated and batches are well-aligned (Supplementary Fig. 2a). We visualized the unshared-bio factors corresponding to each condition type using PCA, and notice that the unshared-bio factors for each condition type effectively separate the conditions within that type, while the batches are still aligned (Supplementary Fig. 2b, c).

In the tests above, we used the simulated datasets where the cell type composition is the same across batches and conditions. In reality, there are challenging scenarios for data integration including: (1) different cell type compositions across batches and conditions, that is, certain cell types exist only in a subset of the data matrices (each data matrix corresponds to a batch and a condition combination); (2) certain cell types have a very small number of cells in some data matrices and are even missing in some other data matrices. These cell types are also known as rare cell types. We conducted a range of simulation tests under the two scenarios above, and confirmed that the shared bio-factors learned by scDisInFact still correctly separate the cell types (Supplementary Figs. 3, 4, 5a). In the case of rare cell types, we generated a simulated dataset of the same structure (Fig. 2c), but with cell type 4 as the rare cell type (Supplementary Fig. 4). Supplementary Fig. 4 shows that in the data matrices before applying scDisInFact, cells from cell type 4 are mixed with other cell types. However, after applying scDisInFact, cells from cell type 4 are located together, making it straightforward to detect this cell type (Supplementary Fig. 5a). Information on the design of the datasets and further analysis of the results are in Supplementary Notes 1 and 2. Given the capability of scDisInFact in integrating datasets with mismatched cell type compositions and detection of rare cell types, it can potentially be used to reveal condition-specific rare cell types which appear to be unaligned cells after integration.

Using the training result from the previous section, we further compared the CKG detection accuracy of scDisInFact and scINSIGHT. We also included Wilcoxon rank sum test (two-sided, not adjusted for multiple comparisons) as an additional baseline method. For each method, we obtain a CKG score for each gene to indicate how likely this gene is a CKG. For scINSIGHT, we used the variance of gene membership matrices across conditions as CKG scores ("Methods"). In Wilcoxon rank sum test, for each gene, we ran the test between the gene's expression levels under different conditions, and obtained the corresponding $p$ values (with false discovery rate multi-tests correction). We transformed the $p$ values into CKG scores following $CKG_{Wilcoxon} = 1 - \frac{p\,val}{\max(p\,val)}$.

AUPRC scores were calculated for each method between the inferred CKG scores and ground truth CKGs ("Methods"). We aggregate the mean AUPRC score of each method on the simulated datasets with different perturbation parameters $\epsilon$ in a barplot (Fig. 2e). scDisInFact consistently performs better than the two baseline methods under all values of the perturbation parameter $\epsilon$, which shows that the successful disentanglement of biological variations and technical batch effect could help to better uncover the CKGs. The AUPRC score increases with the increase of the perturbation parameter. This is

because a higher perturbation parameter corresponds to a larger difference in the expression of the CKGs across conditions, which makes it easier for the CKGs to be detected.

We further investigated scDisInFact's performance in CKG detection using two additional metrics: the Early Precision score (Eprec) and Pearson correlation coefficient between the predicted and ground truth gene scores ("Methods"). The results consistently show the superior performance of scDisInFact against baseline methods (Supplementary Fig. 5b).

We finally test the perturbation prediction accuracy of scDisInFact on the same set of simulated datasets (Fig. 2a). Similar to the generalization test, we conducted perturbation prediction under two different scenarios: (1) In-sample prediction, where the condition to predict is seen in the training dataset, and (2) Out-of-sample prediction, where the condition to predict is not seen in the training dataset.

In the in-sample test, we train scDisInFact using all count matrices in Fig. 2a, and take one count matrix as input to predict the mRNA counts of the same cells under different conditions. For example, in Supplementary Fig. 6a, arrow #2 means that from the data matrix $X_{11}$, predict the cells' expression levels under batch 1, condition <*control*, *severe*>, denoted by $X'_{21}$. We use the notation $X'_{21}$ to distinguish the predicted matrix from $X_{21}$ which is part of the training data. $X'_{21}$ and $X_{21}$ are matrices under the same batch and conditions but on different cells. Therefore, when evaluating the accuracy of $X'_{21}$, it can not be compared with $X_{21}$ at the single cell level, but at the cell type level instead ("Methods").

In the out-of-sample test, we held out all count matrices under condition <*ctrl*, *severe*> ($X_{21}, X_{22}$ in Supplementary Fig. 6b), and trained the model on the remaining count matrices. Then we took one count matrix as input and predicted the corresponding counts under the held-out condition. In the out-of-sample test, <*ctrl, severe*> is the unseen condition because the combination of *ctrl* and *severe* is not seen in the training dataset, although the *ctrl* or *severe* condition can appear in the training data in other condition combinations.

For both the in-sample and out-of-sample predictions, we predict data under condition <*ctrl, severe*> using different matrices as input (Supplementary Fig. 6a, b). Depending on which effects exist between the input and predicted matrices, we categorized the prediction test into 6 scenarios. When the input and predicted matrices are from the same batch, we test the prediction of condition effect of (1) condition Type 1 (treatment, arrow #1 in Supplementary Fig. 6a, b), (2) condition Type 2 (severity, arrow #2 in Supplementary Fig. 6a, b) and (3) condition Types 1 and 2 (arrow #3 in Supplementary Fig. 6a, b). Similarly, when the input and predicted matrices are from different batches, we also test the prediction of condition effect of (4) condition type 1 (arrow #4 in Supplementary Fig. 6a, b), (5) condition type 2 (arrow #5 in Supplementary Fig. 6a, b) and (6) condition type 1 and 2 (arrow #6 in Supplementary Fig. 6a, b). scDisInFact can predict data across any condition and batch combinations, allowing for the prediction of gene expression data for all the cells of all given data matrices under any condition and batch combination.

We compare the performance of scDisInFact with scGen and scPreGAN. As scGen and scPreGAN are only designed for one condition type, we train the methods using the count matrices with fixed condition values for the condition types that we are not predicting ("Methods"). The detailed settings of the training, input, and predicted data matrices for all methods are in Supplementary Tables 2 and 3. We take the known count matrix under the predicted condition and batch, and use this matrix or its denoised matrix as the gold-standard counts ("Methods"). We compare the predicted counts with the gold-standard counts using cell-type-specific MSE, $R^2$, and Pearson correlation scores ("Methods"). The results are summarized in Fig. 3a (in-sample tests) and Fig. 3b (out-of-sample tests). scDisInFact has the smallest MSE and highest Pearson correlation and $R^2$ out of all three methods in all test scenarios. The baseline methods show much higher MSE (note that the

*y*-axis uses log scale in the MSE plots) than scDisInFact while their Pearson correlation can be close to that of scDisInFact. The high MSE of baseline methods is caused by the difference in the distribution between predicted values and ground truth values, although normalization has been performed ("Methods").

Since scDisInFact learns the condition and batch effects from the training dataset, the training dataset with smaller coverage of possible conditions and batches should affect the performance of scDisInFact. We further conducted experiments to analyze how the number of held-out matrices affects the prediction power of scDisInFact. We created 4 scenarios by holding out 1 to 4 count matrices from the training set. The detailed settings of the test scenarios are summarized in Supplementary Table 4. After training the model, we take as input the count matrix corresponding to condition <*stim, severe*> in batch 2 (matrix $X_{42}$ in Supplementary Fig. 6c), and predict the counts under condition <*ctrl, severe*> of batch 1 (matrix $X'_{21}$ in Supplementary Fig. 6c). We measure the cell-type-specific MSE, $R^2$, and Pearson correlation scores (Methods) between the predicted and gold-standard counts (matrices $X_{21}$ and $X'_{21}$) and aggregate the scores into the boxplot in Supplementary Fig. 6d. From Supplementary Fig. 6d, we observe the overall performance of scDisInFact drops when fewer matrices are included in the training data. However, even when holding out 3 matrices, the performance of scDisInFact is comparable or better than the better method out of scGen and scPreGAN for each metric in Fig. 3. With 4 matrices held out (Supplementary Fig. 6c), the performance of scDisInFact deteriorates, as in this case, the difference between any two matrices is a mixture of condition effect and batch effect, which poses a challenging case for disentanglement. In Supplementary Note 3, we discuss the minimum requirements on the input data matrices for perturbation prediction.

## Testing scDisInFact on glioblastoma data
We applied scDisInFact to real datasets. We first applied scDisInFact on a glioblastoma (GBM) dataset[1]. The dataset has 21 count matrices from 6 patient batches with 1 condition type (drug treatment) that includes conditions: no-drug control (*vehicle DMSO*), and panobinostat drug treatment (0.2 uM panobinostat). The metadata of the count matrices follows Supplementary Table 5 and the data matrices can be arranged in a "condition by batch" grid as shown in Supplementary Fig. 8a. Before applying scDisInFact, we filtered the lowly-expressed genes ("Methods") and visualized the dataset using UMAP. Strong technical batch effect and condition effect can be observed among batches and conditions (Supplementary Fig. 8b).

After applying scDisInFact to pre-processed data, we obtained the shared-bio factors and unshared-bio factors that are specific to the two conditions (Fig. 4a and Supplementary Fig. 7c). The shared-bio factors separate cells of major cell types (Fig. 4a left) and are removed of the batch and condition effect (Fig. 4a middle and right). The unshared-bio factors separate cells into different conditions (Supplementary Fig. 7c left), and are removed of the batch effect (Supplementary Fig. 7c right). The latent space visualization shows that scDisInFact is able to disentangle latent factors and remove the strong batch effects in the dataset. We compared the disentanglement results of scDisInFact and scINSIGHT both visually and quantitatively. The calculation of shared-bio and unshared-bio factors of scINSIGHT is described in Methods. We visualized the shared-bio and unshared-bio factors of scINSIGHT using UMAP (Supplementary Fig. 8a). From Supplementary Fig. 8a, the cell types are not well separated in the shared-bio factors (top-left plot) and the batches are not well aligned in unshared-bio factors (bottom-right plot). To quantitatively verify this, we evaluate the shared-bio factor using ASW-batch score and ARI score, where ASW-batch measures the removal of batch effect and ARI measures the separation of cell types ("Methods", Supplementary Fig. 8b). We evaluate the removal of batch effect in unshared-bio factor using ASW-batch scores (Supplementary Fig. 8b). scDisInFact largely outperforms scINSIGHT

in terms of both ARI score for shared-bio factors and ASW-batch score for unshared-bio factors, which is consistent with the visualizations.

We then analyzed the CKGs detected by scDisInFact. After training the model, we obtained the CKG score of each gene from the unshared encoder, and sorted the genes by their scores ("Methods", Fig. 4b). We expect that the top-scoring genes are related to the biological processes associated with panobinostat drug treatment. We obtained high CKG scores for metallothioneins and neuronal marker genes (red dots in Fig. 4b), and macrophage marker genes (blue dots in Fig. 4b)[1]. These top-scoring marker genes are closely related to panobinostat treatment because panobinostat was reported to up-regulate the metallothionein family genes and mature neuronal genes, and down-regulate macrophage marker genes which were immunosuppressive in GBM[1]. We further conducted gene ontology (GO) analysis on the top-300 scoring genes using TopGO[21], and the result shows multiple biological processes relevant to the panobinostat treatment in cancer (Fig. 4c). Panobinostat was shown to activate MHC II pathways[22] and we do observe term that are related to MHC class II pathway proteins. There are also terms related to p53 transcription factor and regulation of growth. P53 is a tumor suppressor involved in the regulation of DNA damage response, which is affected by the use of panobinostat[23]. Panobinostat also was shown to have a growth suppressive effect on cancer[24] (Fig. 4c).

We also tested the perturbation prediction accuracy of scDisInFact on the GBM dataset. We held out the count matrix corresponding to sample "PW034-705" (Supplementary Table 5 and Supplementary Fig. 7a), and trained scDisInFact on the remaining count matrices in the dataset. After training the model, we take one count matrix in the training dataset as input, and predict the counts for the same cells under the condition and batch of the held-out matrix. We choose different count matrices as input to create different scenarios for the prediction task: (1) the input and held-out matrices are from the same batch but under different conditions, where scDisInFact predicts only the condition effect; (2) the input and held-out matrices are from different batches and different conditions, where scDisInFact predicts both the condition effect and batch effect. The configurations of input and output data matrices are in Supplementary Table 6.

We compared the performance of scDisInFact with scGen and scPreGAN for these prediction tasks using cell-type-specific MSE, Pearson correlation, and $R^2$ scores between the predicted and held-out (gold-standard) counts ("Methods"). We calculate the scores for cell types including "Myeloid", "Oligodendrocytes", and "tumor", and aggregate the scores in Fig. 4d. From the comparison result, we observe that scDisInFact has a better prediction accuracy, and the prediction improvement is more pronounced when the batch effect exists between the input and held-out count. This is because scDisInFact specifically models batch effect in the perturbation prediction while the other two methods do not. In both prediction tasks, we further jointly visualize the predicted and the held-out counts using UMAP (Supplementary Fig. 9a for within-batch prediction, Supplementary Fig. 9b for cross-batch prediction). The visualization (Supplementary Fig. 9a, b) shows that the predicted counts of scDisInFact match the held-out counts in both tasks while both baseline methods fail especially in the second task. The visualization result matches the quantitative measurement in Fig. 4d.

## scDisInFact performs disentanglement and predicts data under unseen conditions on COVID-19 dataset
We tested scDisInFact on a COVID-19 dataset with multiple condition types. We built a COVID-19 dataset by collecting data from three different studies[2–4]. Since no significant batch effect is reported within each study[2–4], we treated the data obtained from each study as one batch. The resulting dataset has 3 batches of cells with 2 condition types: age and disease severity. The three batches are termed "Arunachalam_2020", "Lee_2020", and "Wilk_2020" according to the source

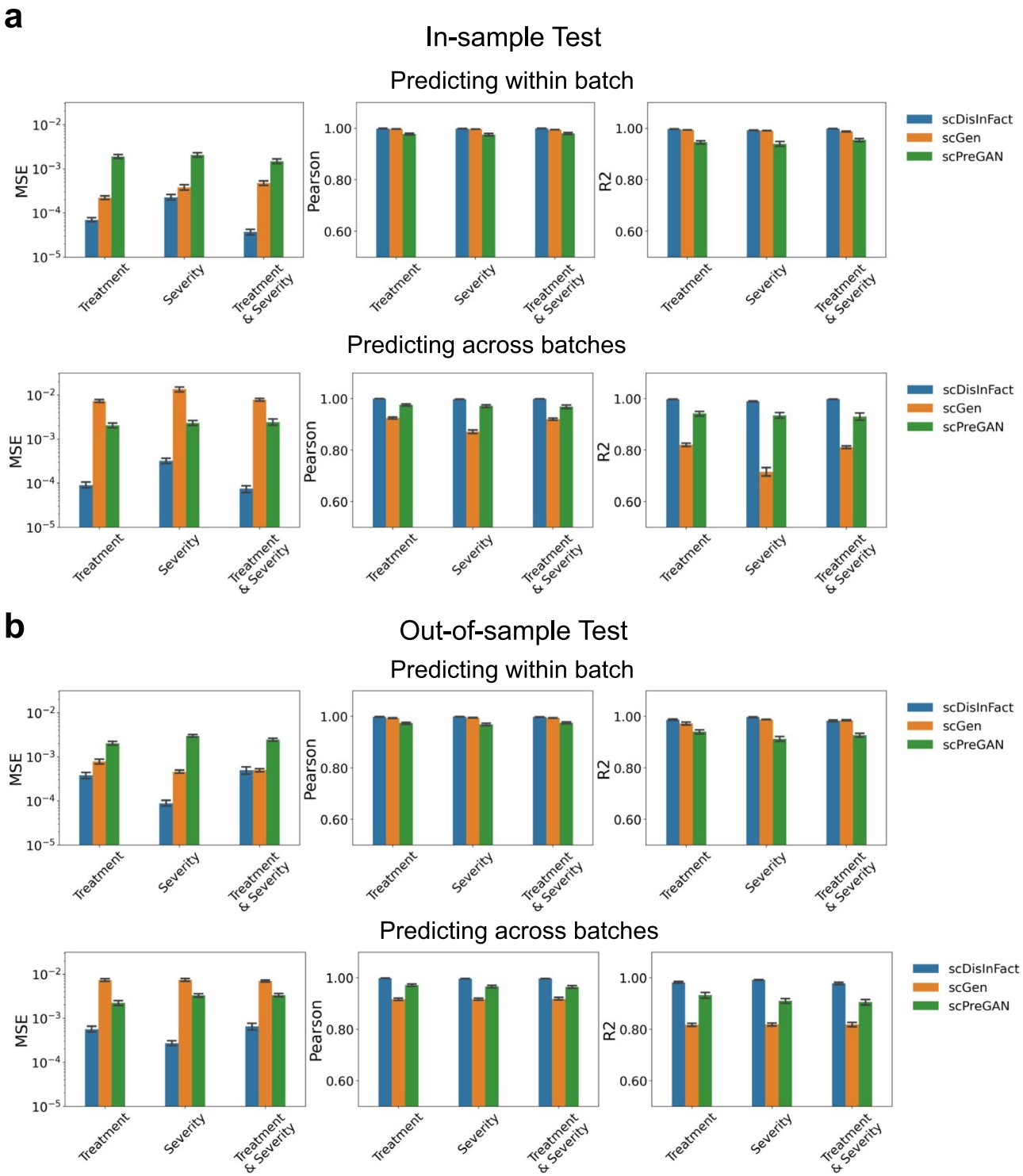

**Fig. 3 | Perturbation prediction results on simulated datasets. a** Perturbation prediction accuracy of scDisInFact and baseline methods in the in-sample test. The accuracy is measured by cell-type-specific MSE (left), Pearson correlation (middle), and $R^2$ scores (right). The upper row shows the result where the input and predicted count matrices are from the same batch, and the lower row shows the result where the input and predicted count matrices are from different batches. Barplots are shown separately for the prediction of "Treatment", "Severity", and "Treatment & Severity" effects. **b** Prediction accuracy of scDisInFact and baseline methods in the out-of-sample test. Barplots are shown separately for the prediction of condition effect ("Treatment", "Severity", "Treatment & Severity") within batch (upper) and across batches (lower). In (**a**) and (**b**), $n = 144$ independent samples are included in each bar. The error bar represents 95% confidence interval, and the center of the error bar shows the mean. Source data for (**a**) and (**b**) are provided in the Source Data file.

of the study. The age condition includes young (40−), middle-aged (40–65), and senior (65+) groups, and the disease severity condition includes healthy control, moderate symptom, and severe symptom ("Methods", the arrangement of count matrices follows Fig. 5a).

We first visualize the gene expression data using UMAP before applying scDisInFact. From the visualization, we observed a strong batch effect among different studies (Supplementary Fig. 10a, b). In addition, the distributions of cells under different conditions also

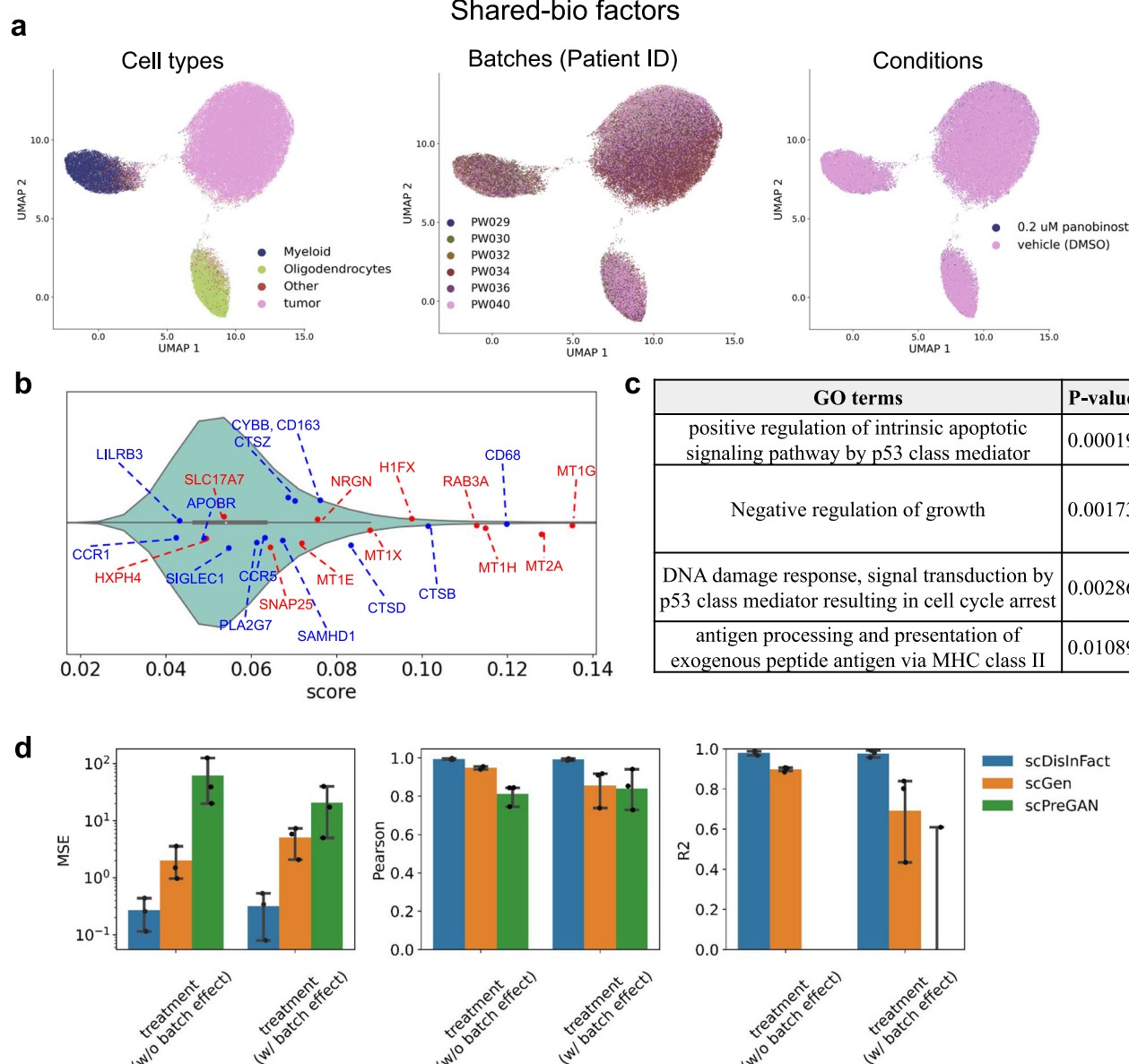

**Fig. 4 | Results on GBM dataset. a** UMAP visualization of shared-bio factors, where the cells are colored by (left) original cell type, (middle) batches, and (right) conditions. **b** Violin plot of the CKG scores learned by scDisInFact. Red dots correspond to metallothioneins and neuronal marker genes, and blue dots correspond to macrophage marker genes. In the violin plot, independent scores of $n = 19{,}949$ genes are included. **c** Top Gene ontology terms inferred from top-scoring genes. $p$ values are calculated from TopGO, which is based on one-sided Fisher's exact test and not adjusted for multiple comparisons. Multiple terms are relevant to Panobinostat treatment, including terms related to MHC class II pathways and p53 regulation. **d** Perturbation prediction accuracy of scDisInFact and baseline methods, which include the cell-type-specific MSE, Pearson correlation, and $R^2$ scores. Barplots are shown separately for the prediction of (1) condition effect within batch ("treatment (w/o batch effect)") and (2) condition effect across batches ("treatment (w/ batch effect)"). In the barplot, $n = 3$ independent samples are included in each bar. The error bar represents 95% confidence interval, and the center of the error bar shows the mean. Source data for (**a**, **b**, and **d**) are provided in the Source Data file.

show variation due to the condition effect (Supplementary Fig. 10c, d). We then trained scDisInFact on the dataset, and visualized the shared and unshared biological factors using UMAP (Fig. 5b, c). The visualization shows that the shared-bio factors are aligned across batches and conditions while maintaining the same level of cell type separation as in individual batches (Fig. 5c and Supplementary Fig. 10a), and the unshared-bio factors are also grouped according to their corresponding condition types (Fig. 5b).

We then tested the perturbation prediction of scDisInFact. Similar to the test on simulated datasets, we design tests separately for the prediction of condition combinations that are seen (in-sample test) and unseen (out-of-sample test) in the training set. In in-sample

test, we trained scDisInFact on all available count matrices (Supplementary Fig. 11a), whereas in out-of-sample test, we held out all count matrices under condition <*moderate, 40–65*> ($X_{51}$ and $X_{52}$ in Supplementary Fig. 11b) and train scDisInFact on the remaining count matrices. In both tests, given an input count matrix, we predict the gene expression data of the same cells under the condition <*moderate, 40–65*> and batch "Lee_2020" ($X'_{52}$ in Supplementary Fig. 8a, b). Depending on whether condition effect and batch effect exist between input and predicted count matrices, we again categorize the prediction tests into 6 scenarios, similar to the test on simulated datasets. When the predicted and input count matrices are from the same batch, we test the prediction of condition effect

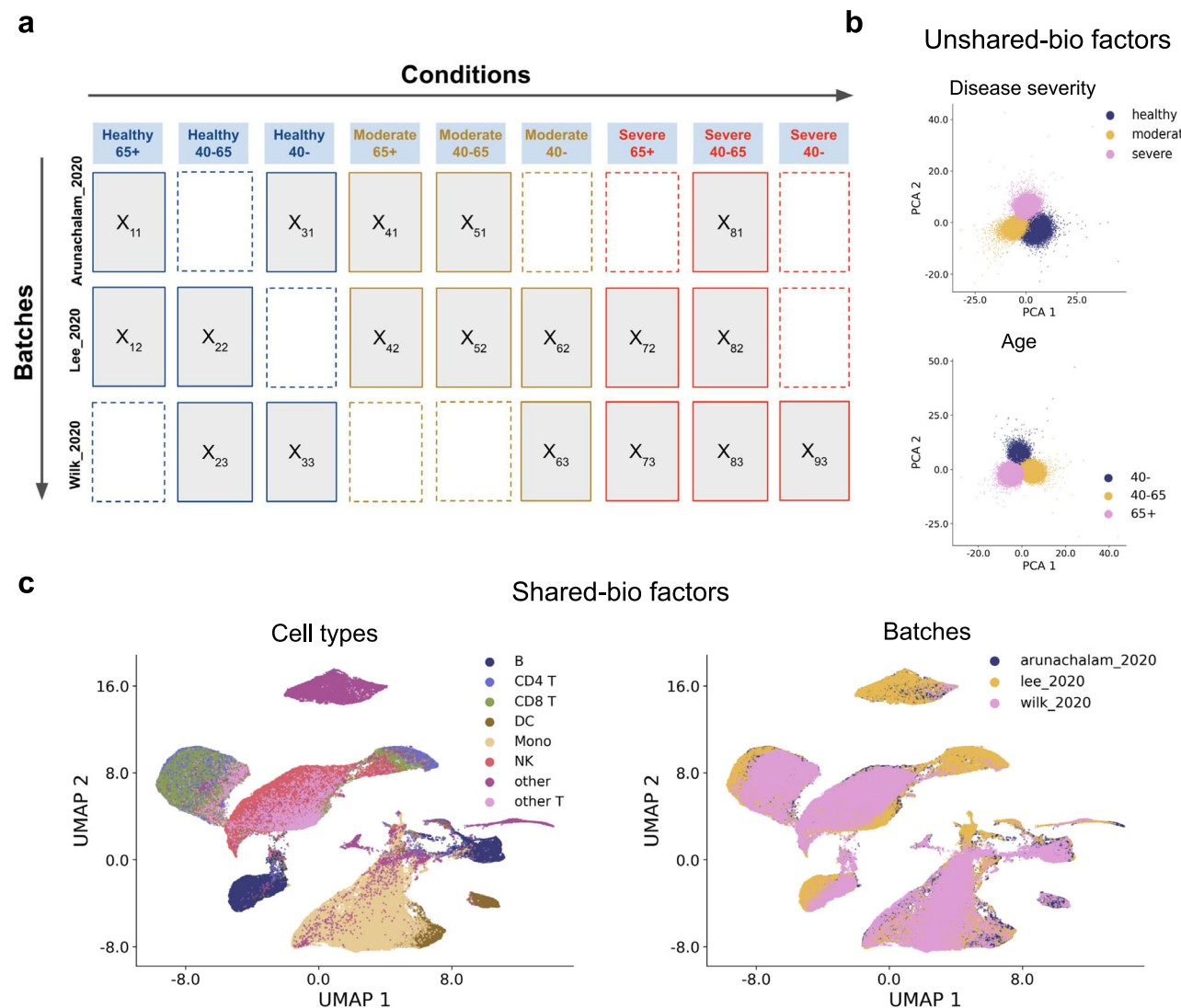

**Fig. 5 | Results on COVID-19 dataset. a** The arrangement of count matrices in the datasets, where matrices are grouped by conditions (columns) and batches (rows). The rectangle with dashed borders means that the count matrix is missing for the corresponding condition and batch. **b** The UMAP visualization of unshared-bio factors in scDisInFact. The upper plot shows the factor that encodes disease severity condition, and the lower plot shows the factor that encodes age condition. Cells are colored according to their ground truth conditions. **c** The UMAP visualization of shared-bio factors in scDisInFact, where cells are colored by ground truth cell types (left) and batches (right). Source data for (**b**) and (**c**) are provided in the Source Data file.

of disease severity (arrows 1 in Supplementary Fig. 8a, b), age (arrows 2 in Supplementary Fig. 8a, b), and disease severity and age (arrows 3 in Supplementary Fig. 11a, b). Similarly, when the predicted and input count matrices are from different batches, we also test the prediction of these three combinations of condition effects (arrows 4, 5, 6 in Supplementary Fig. 11a, b).

Once again, we compared scDisInFact's perturbation prediction performance with scGen and scPreGAN. The detailed settings of the training, input, and predicted conditions for all methods are summarized in Supplementary Table 7. We use the held-out count matrix corresponding to condition *<moderate, 40−65>* and batch "Lee_2020" ($X_{52}$ in Supplementary Fig. 8a, b) as the gold-standard counts for scGen and scPreGAN and its denoised version for scDisInFact, and calculate cell-type-specific MSE, $R^2$, and Pearson correlation scores between predicted and gold-standard counts ("Methods", Fig. 6). In both in-sample test and out-of-sample test, scDisInFact outperforms scGen and scPreGAN across all 6 prediction scenarios, where the improvement scDisInFact brings is more pronounced for the tasks that involve prediction of batch effects, compared to the tasks of predicting only condition effects. This can be attributed to scDisInFact's ability to

model both condition effects and batch effects, whereas scGen and scPreGAN model the condition effect without considering the batch effect.

**Ablation study on the loss terms of scDisInFact**
scDisInFact has several loss terms in its objective function, including the ELBO loss, MMD loss, classification loss, and group-lasso loss (Methods). We validated the effect of each loss term (except for the ELBO loss which is required for the VAE model) through ablation tests, using the same simulated datasets (details in Supplementary Note 4).

We first conducted the ablation test on MMD loss. In scDisInFact, MMD loss is used to ensure the disentanglement of latent factors ("Methods"), which is further expanded into the objectives of (1) removing the batch effect from the shared-bio and unshared-bio factors, (2) removing the condition effect from the shared-bio factors, and (3) disentangling unshared-bio factors of different condition types. We evaluated the effectiveness of MMD loss from these three objectives separately, by measuring the alignment of different latent factors using ASW-batch scores (details in Supplementary Note 4). The results

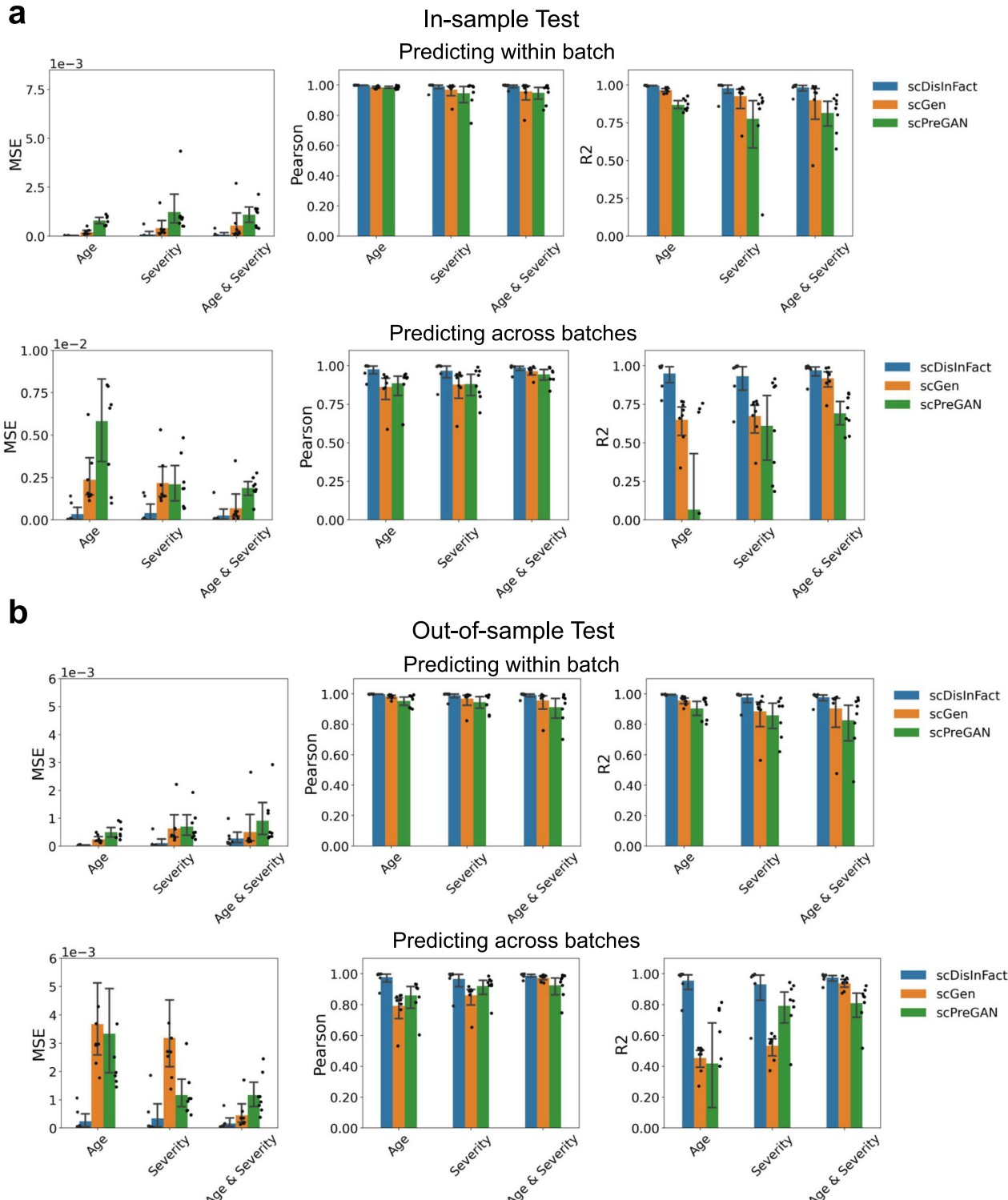

**Fig. 6 | Perturbation prediction result on COVID-19 dataset. a** Perturbation prediction accuracy of scDisInFact and baseline methods in the in-sample test. The accuracy is measured by cell-type-specific MSE (left), Pearson correlation (middle), and $R^2$ scores (right). The upper row shows the result where the input and predicted count matrices are from the same batch, and the lower row shows the result where the input and predicted count matrices are from different batches. Barplots are shown separately for the prediction of "Age", "Severity", and "Age & Severity" effects. **b** Perturbation prediction accuracy of scDisInFact and baseline methods in the out-of-sample test. Barplots are shown separately for the prediction of condition effect ("Age", "Severity", "Age & Severity") within batch (upper) and across batches (lower). In (**a**) and (**b**), $n = 8$ independent samples are included in each bar. The error bar represents 95% confidence interval, and the center of the error bar shows the mean. Source data for (**a**) and (**b**) are provided in the Source Data file.

(Supplementary Fig. 12a–c) confirm the advantage of using MMD loss. MMD loss also affects the task of perturbation prediction. We then compared the perturbation prediction accuracy of the models with and without the MMD loss. We measured the perturbation prediction accuracy using MSE ("Methods"), and the result shows that the use of MMD loss reduces the perturbation prediction errors (Supplementary Fig. 12d).

We then conducted the ablation test on the classification loss. The classification loss separates unshared-bio factors of different conditions, which is essential for the perturbation prediction task. We compared the perturbation prediction accuracy of the models with and without the classification loss. The results (Supplementary Fig. 13) show that the model with classification loss (reg: 1) shows significantly lower MSE compared to the model without classification loss (details in Supplementary Note 4).

We finally conducted the ablation test on group-lasso loss. The group-lasso loss helps the model extract the most important CKGs. We compared the CKGs detection accuracy of the models trained with and without group-lasso loss (measured by AUPRC scores). As expected, the use of group-lasso loss (reg: 1) significantly improves the CKGs detection accuracy of the model (Supplementary Fig. 13, details in Supplementary Note 4).

### Hyper-parameter test of scDisInFact

We evaluated the model performance under different hyper-parameter settings. The main hyper-parameters of the model include the regularization weights ($\lambda_{kl}^1$, $\lambda_{kl}^2$, $\lambda_{gl}$, $\lambda_{mmd}$, $\lambda_{ce}$) and the number of latent dimensions (notations introduced in "Methods"). scDisInFact performs multiple tasks, thus there is no unified way to evaluate the overall performance of the model. Here we narrow down the performance evaluation into perturbation prediction on held-out condition (measured by MSE), and CKGs detection (measured by AUPRC scores). Perturbation prediction on held-out condition measures the overall generalization ability of scDisInFact, which relies on the quality of the learned latent factors. CKGs detection is a standalone task that needs to be measured independently. We use the simulated datasets for the hyper-parameter test (design of the test in Supplementary Note 5), and the test result is shown in Supplementary Fig. 14.

The test result shows that the perturbation prediction accuracy increases with the increase of MMD weight $\lambda_{mmd}$, whereas the CKGs detection accuracy decreases with the increase of MMD weight. MMD weight that is too large makes it hard to deal with cell type mismatch across batches and conditions. $\lambda_{mmd} = 10^{-4}$ shows a more balanced result. The CKGs detection accuracy of scDisInFact increases with the increase of group-lasso weight $\lambda_{gl}$, and the perturbation prediction achieves the highest accuracy when $\lambda_{gl} = 1$. Classification weight $\lambda_{ce}$ around 0.1 - 1 shows the best performance in both perturbation prediction and CKGs detection. We fix the KL divergence weight of the shared-bio factors to be a small value ($10^{-5}$) to encourage the separation of cell types in the latent space, and test the KL divergence weight of the unshared-bio factors. $\lambda_{kl}^2 = 10^{-4} \sim 10^{-2}$ shows the best performance. The overall performance of the model is not affected significantly by the latent dimensions (the number of shared dimensions between 4 and 16, unshared dimensions between 2 and 4). The hyper-parameter tests show that scDisInFact has the best performance near the recommended hyper-parameter settings ($\lambda_{kl}^1 = 10^{-5}$, $\lambda_{kl}^2 = 10^{-2}$, $\lambda_{mmd} = 10^{-4}$, $\lambda_{ce} = 1$, $\lambda_{gl} = 1$, "Methods").

### Running time comparison of scDisInFact and baseline methods

We measured the running time of scDisInFact on simulated datasets of different sizes along with baseline methods, including scGEN, scPreGAN, and scINSIGHT. Supplementary Fig. 15 shows that scDisInFact scales well to large datasets, similar to scGEN and scPreGAN. These methods are faster than scINSIGHT thanks to their usage of GPU to accelerate the training.

## Discussion

We presented scDisInFact, a deep learning framework that models multi-batch multi-condition scRNA-seq datasets. scDisInFact is a unified framework for three prominent tasks in disease studies: (1) the disentanglement of biological factors and removal of batch effect, (2) detection of condition-associated key genes, and (3) prediction of gene expression data under conditions where no data is measured. The last two tasks are enabled by achieving the goal of the first task, the disentanglement of variations in a multi-batch multi-condition scRNA-seq dataset. While these tasks have been studied separately in existing literature, it is important to recognize that they are related to each other and can be addressed under a unified framework. scDisInFact performs a comprehensive disentanglement of various types of information in the dataset, which enables its multi-task ability.

The extensive tests conducted on simulated and real datasets support that scDisInFact has superior performance than the baseline methods that can only conduct one task. Not only scDisInFact gains better performance on these tasks, but it is also more versatile than existing methods in each task which can lead to broader applications under realistic scenarios. For batch effect removal, scDisInFact removes only batch effects and preserves biological differences across data matrices. For condition-associated key gene detection, not only scDisInFact can output CKGs at a high level, but the perturbation prediction results can also be used to find genes that are differentially expressed in specific cells or cell types from one condition combination to any other condition combination. For perturbation prediction, scDisInFact models multiple condition types associated with the donors and can predict data from a condition combination to any other combination under study. This enables applications in complex scenarios like predicting the effect of combinations of multiple drugs.

While scDisInFact performs well in the reported scenarios which are applicable to a wide range of real datasets, there are certain scenarios that can pose challenges to disentanglement methods. As shown in our results, when each batch of cells is measured under only one condition, there is not enough information to fully disentangle the batch and condition effect. Measuring data from multiple conditions in the same batch can largely ameliorate this problem, which echoes recommendations on wet-lab experimental design that batches should be distributed across biological conditions[25,26].

In order to be equipped with new features that were not provided in existing related work (e.g. disentangling unshared bio-factors for multiple condition types), scDisInFact made a few assumptions about the multi-batch multi-condition datasets. First, scDisInFact assumes that the condition types are mutually independent. Such independence assumptions are often used in disentanglement learning methods[27,28], and are needed for perturbation prediction under multiple condition types. However, we acknowledge that such independence does not always hold for real-world datasets. In our tests, we selected condition types that showed little correlation. For the condition types that show a strong correlation, we suggest users combine them into a joint condition type instead of treating them separately. For example, given a dataset with condition type 1 (stim1, ctrl1) and condition type 2 (stim2, ctrl2) that are highly correlated, the users can create a combined condition type (stim1_stim2, ctrl1_stim2, stim1_ctrl2, ctrl1_ctrl2) and train scDisInFact on the combined condition type instead. scDisInFact also assumes that the condition effect is independent of the batch effect, and batch effects are consistent across conditions. Such an assumption is made because the source of the batch effect is technical confounders in the experiment whereas the source of the condition effect is purely biological. In perturbation prediction, following previous work[17,29], scDisInFact assumes the condition and batch effects are additive to the cells' original biological identity (shared-bio factors) in the latent space. Convenient as it is in constructing models, this assumption also has its limitations in modeling higher-order productive conditions or batch effects. Future

works should focus on better accommodating the correlation between condition effects and batch effects, and further improve the modeling capacity of the perturbation effect.

## Methods

### Loss function of scDisInFact

scDisInFact uses a combination of loss terms to accomplish the given tasks. Firstly, scDisInFact reconstructs the input gene expression data from the decoder by minimizing the evidence lower bound (ELBO) loss, following the design of variational autoencoder. We denote the input gene expression data, shared-bio factors, unshared-bio factors, and batch factors of each cell as $\mathbf{x}$, $\mathbf{z}_s$, $\mathbf{z}_u$, and $b$, respectively. There can be multiple unshared-bio factors $\mathbf{z}_u$s, each corresponding to one condition type (Fig. 1b). For clarity of explanation, we describe the case where only one condition type exists in the Method section, and the multiple-condition-type case can be easily generalized from one condition type. The ELBO loss for datasets with one condition type follows:

$$L_{\text{elbo}}(\mathbf{\Theta},\mathbf{\Phi}) = -\mathbb{E}_{\mathbf{z}_s,\mathbf{z}_u,b}[\log P_{\mathbf{\Phi}}(\mathbf{x}|\mathbf{z}_s,\mathbf{z}_u,b)] + \text{KL}(Q_{\mathbf{\Theta}}(\mathbf{z}_s,\mathbf{z}_u|b,\mathbf{x}) \parallel P(\mathbf{z}_s,\mathbf{z}_u|b)) \tag{1}$$

$Q_{\mathbf{\Theta}}(\cdot)$ is the encoder network with parameters $\mathbf{\Theta}$ that model the posterior distribution of $\mathbf{z}_s$ and $\mathbf{z}_u$ given $\mathbf{x}$ and $b$. $P_{\mathbf{\Phi}}(\cdot)$ is the decoder network with parameter $\mathbf{\Phi}$ that models the likelihood function of $\mathbf{x}$. $P(\mathbf{z}_s, \mathbf{z}_u|b)$ is the prior distribution of $\mathbf{z}_s$ and $\mathbf{z}_u$. Since $\mathbf{z}_s$, $\mathbf{z}_u$, and $b$ model factors that correspond to independent sources of variation, we can factorize $Q_{\mathbf{\Theta}}(\mathbf{z}_s, \mathbf{z}_u|b, \mathbf{x})$ into $Q_{\mathbf{\Theta}}(\mathbf{z}_s|\mathbf{x}, b)Q_{\mathbf{\Theta}}(\mathbf{z}_u|\mathbf{x}, b)$, and $P(\mathbf{z}_s, \mathbf{z}_u|b)$ into $P(\mathbf{z}_s|b)P(\mathbf{z}_u|b)$ according to mean-field approximation[30]. We use a shared encoder to model $Q_{\mathbf{\Theta}}(\mathbf{z}_s|\mathbf{x}, b)$, and unshared encoders to model $Q_{\mathbf{\Theta}}(\mathbf{z}_u|\mathbf{x})$ ($b$ is not used in the input of the unshared encoder). Then the ELBO loss can be simplified as:

$$L_{\text{elbo}}(\mathbf{\Theta},\mathbf{\Phi}) = -\mathbb{E}_{\mathbf{z}_s,\mathbf{z}_u,b}[\log P_{\mathbf{\Phi}}(\mathbf{x}|\mathbf{z}_s,\mathbf{z}_u,b)] + \text{KL}(Q_{\mathbf{\Theta}}(\mathbf{z}_s|\mathbf{x},b) \parallel P(\mathbf{z}_s|b))$$
$$+ \text{KL}(Q_{\mathbf{\Theta}}(\mathbf{z}_u|\mathbf{x}) \parallel P(\mathbf{z}_u|b)) \tag{2}$$

We model the likelihood function $P_{\mathbf{\Phi}}(\mathbf{x}|\mathbf{z}_s, \mathbf{z}_u, b)$ using a negative binomial distribution:

$$\text{NB}(\mathbf{x};\mu,\theta) = \frac{\Gamma(\mathbf{x}+\theta)}{\Gamma(\theta)}\left(\frac{\theta}{\theta+\mu}\right)^{\theta}\left(\frac{\mu}{\theta+\mu}\right)^{\mathbf{x}} \tag{3}$$

where $\mu$ and $\theta$ are the mean and dispersion parameters of the distribution[31,32]. The decoder learns the gene-specific $\mu$ and $\theta$ for each cell. By minimizing ELBO loss, scDisInFact is trained to extract the main biological variation in the dataset into latent factors and generate correct gene expression data from latent factors.

In Eq. (2), the prior distributions $P(\mathbf{z}_u|b)$ and $P(\mathbf{z}_s|b)$ remains to be decided. For the shared-bio factors, we use a standard normal distribution as the prior following the classical VAE model[19], which means $P(\mathbf{z}_s|b) = P(\mathbf{z}_s) = N(\mathbf{0}, \mathbf{I})$. For the unshared-bio factors, we use a mixture of Gaussian prior since we already know their condition labels as the prior information. In the Gaussian mixture prior, we have cells of each condition label following one Gaussian distribution with learnable mean and variance. The total number of Gaussian distributions within the mixture prior is decided by the number of unique condition labels within the data. Given unshared-bio factors $\mathbf{z}_s$ from condition $c$ and batch $b$, the prior distribution is written as $P(\mathbf{z}_u|b) = P(\mathbf{z}_u|c) = N(\boldsymbol{\mu}_c, \boldsymbol{\sigma}_c\mathbf{I})$, where $\boldsymbol{\mu}_c$ and $\boldsymbol{\sigma}_c$ are learned using stochastic gradient descent along with the other model parameters.

When the condition effect is not strong enough, the prior parameters $\boldsymbol{\mu}_c$ and $\boldsymbol{\sigma}_c$ of different condition label $c$s tend to "collapse" into the same value. To better separate the unshared-bio factors of different condition types, we apply a linear classifier on the unshared-bio factors and use it to predict the condition label of each cell. The unshared encoder is trained jointly with the attached classifier for condition label prediction. This is to help the unshared encoder extract the biological information that is important to the condition effect. We measure the prediction accuracy using cross-entropy loss $L_{\text{ce}}(y_{\text{cond}}, \hat{y}_{\text{cond}})$, where $\hat{y}_{\text{cond}}$ is the classifier output and $y_{\text{cond}}$ is the condition label.

scDisInFact further applies maximum mean discrepancy (MMD) loss on $\mathbf{z}_s$ and $\mathbf{z}_u$ to ensure the disentanglement of these factors. The MMD loss calculates the degree of mismatch between two distributions, and was used to align the latent embedding of cells across batches[11,33] (Supplementary Note 6). In scDisInFact, MMD loss is applied on $\mathbf{z}_s$ to ensure that $\mathbf{z}_s$ is independent of condition (including all condition types) and batch, which means $\mathbf{z}_s$ should follow the same distribution regardless of the condition label $c$ and batch $b$ it belongs to. Then MMD loss is applied on $\mathbf{z}_s$ across batches and conditions. We denote the set of batches under condition label $c$ as $B_c$, and the total number of condition labels as $C$ (including all condition types). The MMD loss on $\mathbf{z}_s$ is the sum of MMD losses between $\mathbf{z}_s$ from a reference batch and condition ($\mathbf{z}_s^{\text{ref}}$) and $\mathbf{z}_s$ from each of the remaining batches and conditions ($\mathbf{z}_s^i$):

$$L_{\text{mmd}}(\mathbf{z}_s) = \sum_{c=1}^{C}\sum_{i \in B_c} L_{\text{mmd}}(\mathbf{z}_s^{\text{ref}},\mathbf{z}_s^i) \tag{4}$$

We used the $\mathbf{z}_s$ of the batch and condition label with the largest cell population as $\mathbf{z}_s^{\text{ref}}$ in all our tests.

We then consider the MMD loss that is applied on $\mathbf{z}_u$. The definition of MMD loss on $\mathbf{z}_u$ is slightly different between the cases when the dataset has one condition type and multiple condition types. We first consider the case where there is only one condition type in the dataset. In this case, MMD loss is used to ensure that $\mathbf{z}_u$ is independent of batch $b$. For the cells of each condition label $c$, their corresponding unshared-bio factors $\mathbf{z}_u$ should be independent of batch $b$, which means the $\mathbf{z}_u$ of different batches $b$ should follow the same distribution. On the contrary, the $\mathbf{z}_u$ of different condition labels are no longer required to follow the same distribution. As a result, the MMD loss should be applied separately on $\mathbf{z}_u$ of each condition labels. The final MMD loss follows:

$$L_{\text{mmd}}(\mathbf{z}_u) = \sum_{c=1}^{C}\sum_{i \in B_c} L_{\text{mmd}}(\mathbf{z}_u^{\text{ref}(c)},\mathbf{z}_u^i) \tag{5}$$

ref($c$) is the reference batch under condition label $c$, and it is selected to be the largest batch of the data corresponding to condition label $c$. When there is more than one condition type, MMD loss is used to ensure that $\mathbf{z}_u$ of each condition type is independent of both the batch and the other irrelevant condition types. In this case, different condition labels of the irrelevant condition types should also be considered as different "batches" when formulating MMD loss of $\mathbf{z}_u$. The MMD loss still follows Eq. (5), but the definition of batch set in $B_c$ should be expanded to all unique combinations of batches and condition labels in irrelevant condition types.

We transform the first layer of each unshared encoder into a feature (gene) selection layer through group lasso loss[34,35]. We represent the weight matrix of the first layer by $\mathbf{W} = [\mathbf{w}_1, \mathbf{w}_2, \cdots, \mathbf{w}_d]$, where $d$ is the number of input dimensions (genes), and $\mathbf{w}_i$ is the $i$th column vector of $\mathbf{W}$ connecting to the $i$th gene. The group lasso loss $L_{gl}(\mathbf{W}) = \sum_{i=1}^{d} \parallel \mathbf{w}_i \parallel_2$ penalizes the number of non-zero $\mathbf{w}_i$s, thus the first layer of each unshared encoder is forced to select the most discriminative genes as the condition-associated key genes (CKGs) of the corresponding condition type.

The objective function of scDisInFact consists of a weighted combination of losses above, which follows:

$$\min_{\Theta,\Phi} -\mathbb{E}_{\mathbf{z}_s,\mathbf{z}_u,b}[\log P_{\Phi}(\mathbf{x}|\mathbf{z}_s,\mathbf{z}_u,b)] + \lambda_{\mathrm{kl}}^1 \mathrm{KL}(Q_{\Theta}(\mathbf{z}_s|\mathbf{x},b) \parallel P(\mathbf{z}_s))$$
$$+ \lambda_{\mathrm{kl}}^2 \mathrm{KL}(Q_{\Theta}(\mathbf{z}_u|\mathbf{x},b) \parallel P(\mathbf{z}_u|c)) + \lambda_{\mathrm{gl}} L_{gl}(\mathbf{W}) \qquad (6)$$
$$+ \lambda_{\mathrm{mmd}}(L_{\mathrm{mmd}}(\mathbf{z}_s) + L_{\mathrm{mmd}}(\mathbf{z}_u)) + \lambda_{\mathrm{ce}} L_{\mathrm{ce}}(y_{\mathrm{cond}}, \hat{y}_{\mathrm{cond}})$$

where $\lambda_{\mathrm{kl}}^1$, $\lambda_{\mathrm{kl}}^2$, $\lambda_{\mathrm{ce}}$, $\lambda_{\mathrm{mmd}}$, and $\lambda_{\mathrm{gl}}$ are the weights of the losses.

## Training algorithm of scDisInFact

We update the model parameter in an alternating manner using stochastic gradient descent. For each iteration, the parameter update of scDisInFact is separated into two steps. We first fix the parameters of the unshared encoder and classifier and update the parameters of the shared encoder and decoder through stochastic gradient descent. The loss function (Eq. (6)) is then simplified into:

$$L_{s1} = -\mathbb{E}_{\mathbf{z}_s,\mathbf{z}_u,b}[\log P_{\Phi}(\mathbf{x}|\mathbf{z}_s,\mathbf{z}_u,b)] + \lambda_{\mathrm{kl}}^1 \mathrm{KL}(Q_{\Theta}(\mathbf{z}_s|\mathbf{x},b) \parallel P(\mathbf{z}_s)) + \lambda_{\mathrm{mmd}} L_{\mathrm{mmd}}(\mathbf{z}_s)$$
$$(7)$$

Then we fix the parameters of the shared encoder, and update the parameters of the unshared encoder, classifier, and decoder through stochastic gradient descent. The loss function (Eq. (6)) is then simplified into:

$$L_{s2} = -\mathbb{E}_{\mathbf{z}_s,\mathbf{z}_u,b}[\log P_{\Phi}(\mathbf{x}|\mathbf{z}_s,\mathbf{z}_u,b)] + \lambda_{\mathrm{kl}}^2 \mathrm{KL}(Q_{\Theta}(\mathbf{z}_u|\mathbf{x},b) \parallel P(\mathbf{z}_u)) + \lambda_{\mathrm{mmd}} L_{\mathrm{mmd}}(\mathbf{z}_u)$$
$$+ \lambda_{\mathrm{ce}} L_{\mathrm{ce}}(y_{\mathrm{cond}}, \hat{y}_{\mathrm{cond}}) + \lambda_{\mathrm{gl}} L_{gl}(\mathbf{W})$$
$$(8)$$

The algorithm iterates until the objective function (Eq. (6)) converges. We trained scDisInFact using Adam optimizer[36], and set the learning rate to be $5 \times 10^{-4}$ and the batch size to be 64. Supplementary Fig. 16 shows the training loss curves of scDisInFact on all testing datasets in the manuscript.

## Condition-associated key gene (CKG) detection

The weight matrix $\mathbf{W}$ in the first layer of each unshared encoder is used to extract the CKGs of its corresponding condition type (Supplementary Fig. 1a). Each column vector of $\mathbf{W}$ is connected to one input gene, and we used the $\ell_2$-norm of each column vector as the score of the corresponding gene. For gene $i$, the corresponding score $s_i$ is calculated as $s_i = \|\mathbf{w}_i\|_2$, where $\mathbf{w}_i$ is the $i$th column vector of $\mathbf{W}$. A higher $s_i$ score means that gene $i$ is more likely to be a CKG.

## Prediction of condition effect on gene expression data

Given input gene expression data under one condition, scDisInFact is able to predict the corresponding data under other conditions. We illustrate the prediction procedure of scDisInFact using the example in Supplementary Fig. 1b. Supplementary Fig. 1b describes a case where the dataset has 3 condition labels (condition 1, 2, and 3 in unshared-bio factors) and 6 batches (6 dimensions in batch factors). In the example, scDisInFact takes as input a count matrix under condition 1 and batch 6, and predicts the count under condition 2 and batch 1 (Supplementary Fig. 1b). To do the prediction, we need to calculate new unshared-bio factors through latent space arithmetics[17] (Supplementary Fig. 1b (left)). The latent space arithmetics includes two steps: (1) we calculate the shifting vector $\boldsymbol{\delta}$ that measures the difference between the mean unshared-bio factors under condition 1 ($\bar{\mathbf{z}}_u^1$) and condition 2 ($\bar{\mathbf{z}}_u^2$), following $\boldsymbol{\delta} = \bar{\mathbf{z}}_u^2 - \bar{\mathbf{z}}_u^1$; (2) For the input cells (under condition 1), we shift their unshared-bio factors $\mathbf{z}_u^1$ by $\boldsymbol{\delta}$, following $\mathbf{z}_u^{1'} = \mathbf{z}_u^1 + \boldsymbol{\delta}$. Then we need to update the batch factor to match the predicted batch. In Supplementary Fig. 1b, since the predicted count belongs to batch 1, we assign 1 to the 1st dimension of the batch factor and 0s to the remaining dimensions. We keep the shared-bio factors the same

since the shared-bio factors do not encode any condition or batch effect. Finally, we feed the original shared-bio factors, the updated unshared-bio factors, and the updated batch factors into the decoder, and use the decoder output $\boldsymbol{\mu}$ as the predicted counts. Decoder models the gene expression data using a negative binomial distribution, and the output $\boldsymbol{\mu}$ is interpreted as the denoised gene expression data.

## Hyper-parameter selection

The main hyper-parameters of scDisInFact include the latent dimensions of shared-bio and unshared-bio factors, and the weight parameters in the loss function. The most common way of hyper-parameter selection on such a model is conducting a grid search of the hyper-parameter on the held-out dataset. Given a dataset, one can separate 10% of cells from each batch to create a testing dataset, train the model on the remaining dataset, and check the losses on the testing dataset (e.g., log-likelihood loss, classification loss, etc.). However, the grid search is extremely time-consuming given a large set of hyper-parameters or a large input dataset. Here we also provide a recommended hyper-parameter setting, which was obtained from extensive tests on both real and simulated datasets. We recommend the latent dimensions of the shared encoder to be 8 and of unshared encoders to be 2. The recommended weights are $\lambda_{\mathrm{kl}}^1 = 10^{-5}$, $\lambda_{\mathrm{kl}}^2 = 10^{-2}$, $\lambda_{\mathrm{mmd}} = 10^{-4}$, $\lambda_{\mathrm{ce}} = 1$, and $\lambda_{\mathrm{gl}} = 1$. We used the recommended hyper-parameters for all our test results in the manuscript. Users can manually tune the hyper-parameters around the recommended setting, and it should provide a reasonable result on most of the dataset. The latent dimension should be set according to the complexity of the data (cell types, trajectories, condition labels, etc.), where datasets with more complex structures should in general have higher latent dimensions. The regularization weights control the effect of loss terms in contributing to the final results. $\lambda_{\mathrm{kl}}^1$ and $\lambda_{\mathrm{kl}}^2$ controls how close the shared-bio factors and the unshared-bio factors are to the prior. A higher $\lambda_{\mathrm{kl}}^1$ makes the shared-bio factors to be more close to the Gaussian prior. A higher $\lambda_{\mathrm{kl}}^2$ makes the unshared-bio factors to be more close to the Gaussian mixture prior. $\lambda_{\mathrm{mmd}}$ controls the independence of different factors and conditions/batches. A higher $\lambda_{\mathrm{mmd}}$ better disentangle different latent factors and batch effect, but a value that is too high will cause overcorrection of batch and condition effect of the shared-bio factors. $\lambda_{\mathrm{ce}}$ controls the classification accuracy, and a higher $\lambda_{\mathrm{ce}}$ makes the unshared-bio factors of different condition labels more separated in the latent space. $\lambda_{\mathrm{gl}}$ controls the effectiveness of CKGs detection in the model, a higher $\lambda_{\mathrm{gl}}$ makes the model weight more on CKGs detection. The final result is a joint contribution of all terms, and any term with too high a regularization weight will reduce the effectiveness of the remaining terms. It is important to keep all the regularization weights within a reasonable range. The users should also make sure that the visualization of shared-bio and unshared-bio factors always show reasonable results during the tunning: the visualization of the shared-bio factors should have clear cell type separation patterns and be removed of both batch and condition effects; the visualization of the unshared-bio factors should separate the conditions of different condition labels and be removed of batch effect.

We also provide the detailed parameter of neural networks in scDisInFact (network diagram in Supplementary Table 9). The shared encoder is a 3-layer fully connected neural network. The first 2 layers have the same structure, where each layer consists of a linear layer followed by a ReLu activation function and a dropout layer ("linear"-"ReLu"-"dropout", 128 output neurons for each layer). The last layer has two linear networks that produce the mean and variance of the shared-bio factors separately. The unshared encoder has 2 layers. The first layer also follows the "linear"-"ReLu"-"dropout" structure (128 output neurons), and the last layer has two separate linear networks for the mean and variance of the unshared-bio factors. The decoder is a 3-layer fully connected neural network. The first 2 layers also follow the

"linear"-"ReLu"-"Dropout" structure (128 output neurons for each layer). The last layer includes two linear networks that separately produce the mean and dispersion parameters of the data distribution. The dropout rate of all dropout layers in scDisInFact is 0.2.

## Simulation procedure

We simulated multi-batch scRNA-seq datasets using SymSim[37], and then added condition effect on the simulated dataset. We selected $m_{\text{diff}}^c$ genes as the CKGs for each condition type $c$ (CKGs of different condition types have no overlap), and added the condition effect of condition type $c$ on the corresponding CKGs in the Symsim-generated data. We denote the Symsim-generated count matrix as $\mathbf{X}_{\text{obs}}$, where $\mathbf{X}_{\text{obs}}[i,g]$ corresponds to the expression level of a CKG $g$ in cell $i$. Then the condition effect was added as a uniform distribution on $\mathbf{X}_{\text{obs}}[i,g]$ with lower bound $\epsilon - 1$ and upper bound $\epsilon$: $\mathbf{X}'_{\text{obs}}[i,g] = \mathbf{X}_{\text{obs}}[i,g] + \text{unif}(\epsilon - 1, \epsilon)$. $\epsilon$ is the perturbation parameter that controls the strength of the condition effect. For each condition type, multiple conditions can be generated with different condition labels. For example, one can generate count matrices under the control and stimulated conditions, where no condition effect is added on the control condition, and the condition effect on the count matrices in the stimulated condition is added following $\mathbf{X}'_{\text{obs}}[i,g] = \mathbf{X}_{\text{obs}}[i,g] + \text{unif}(\epsilon - 1, \epsilon)$. The modeling of batch effect is already included in Symsim simulator[37].

We generated 9 simulated datasets with 2 condition types and 2 data batches (8 count matrices for each dataset, Fig. 2a). The 2 condition types respectively have condition labels (1) control and stimulation, (2) healthy and severe. We set the first $m_{\text{diff}}^c$ genes to be the CKGs of conditions control and stimulation. We set the $(m_{\text{diff}}^c + 1)$th to $2m_{\text{diff}}^c$th genes to be CKGs of conditions healthy and severe. We first generate 2 batches of count matrices using Symsim. Then we evenly separate cells of each batch and the corresponding gene expression data into 4 conditions: (control, healthy), (control, severe), (stimulation, healthy), and (stimulation, severe). For each chunk of the gene expression data, we add condition effect according to its condition group. We do not add condition effect for the gene expression data under (control, healthy), whereas we add condition effect to the corresponding CKGs for gene expression data under either stimulated or severe conditions. The 9 datasets are generated with different simulation parameters to model different strengths of condition effect. We used 3 CKG numbers ($m_{\text{diff}}^c = 20$, 50, and 100), and generated 3 datasets under each CKG number with 3 perturbation parameters ($\epsilon = 2, 4$ and 8). The detailed simulation parameter settings of 9 datasets are included in Supplementary Table 1.

## Pre-processing steps of real datasets

scDisInFact can be trained directly on the raw scRNA-seq dataset, no pre-processing step is required before running scDisInFact. However, directly training on raw scRNA-seq data can be time-consuming as the feature dimension (number of genes) in the raw data can be extremely high (20,000 - 30,000). Some additional gene filtering steps that remove genes with low expression levels are also recommended, as they can greatly improve the running speed of the model.

## Pre-processing steps of GBM dataset.

In the GBM dataset, the authors obtained multi-batch scRNA-seq data from the GBM resections of patients with different drug treatments. We selected 21 count matrices from 6 GBM patient batches with and without panobinostat drug treatment (respectively named 0.2 uM panobinostat and vehicle (DMSO)), where 16 matrices were under vehicle (DMSO) condition and 5 matrices were under 0.2 uM panobinostat condition (see Supplementary Table 5 for detailed information on the selected batches). We remove the genes that have counts in less than 100 cells across all batches, and obtained count matrices with 74,777 cells and 19,949 genes. The original paper annotated the cells into tumor and non-tumor cells, which is a very high-level annotation. We further annotated the non-tumor cells into Myeloid, Oligodendrocytes, and Other cells using their marker genes[1] for each batch separately.

## Pre-processing steps of COVID-19 dataset.

We selected COVID-19 scRNA-seq studies from the recent summary paper[16] (data downloaded from https://atlas.fredhutch.org/fredhutch/covid/), and used the count matrices stored in "arunachalam_2020_processed.HDF5", "wilk_2020_processed.HDF5", and "lee_2020_processed.HDF5". We followed its standard of disease severity classification, and select the data under the condition "healthy", "moderate", and "severe". We categorized the ages of patients in the studies into groups of "40−" (below 40), "40−65" (between 40 and 45), and "65+" (above 65). Then, we selected the genes that are shared among all three studies. We did not conduct further filtering steps of genes and cells in these studies.

## Details on running baseline methods

**Running scINSIGHT.** When running scINSIGHT on simulated datasets, we preprocessed the dataset following the Seurat normalization steps (as was recommended in scINSIGHT online tutorial: https://github.com/Vivianstats/scINSIGHT/wiki/scINSIGHT-vignette) and did not conduct gene filtering steps on the datasets. The main hyperparameters of scINSIGHT include the number of common gene modules ($K$) and the number of condition-specific gene modules ($K_j$). We used the default setting of $K$, where scINSIGHT searched through $K = 5, 7, 9, 11, 13, 15$ and selected the best performing $K$.

When using scINSIGHT to learn shared-bio and unshared-bio factors, we used the default setting of $K_j$ ($K_j = 2$) and remaining hyperparameters. scINSIGHT learns a factorized common module and a factorized condition-specific module. The common module includes a membership matrix $\mathbf{V}$ and expression levels matrices $\mathbf{W}_{b2}$ ($b = 1, 2, \cdots, B$ for $B$ batches). $\mathbf{W}_{b2}$ is equivalent to the shared-bio factors of cells under batch $b$. The condition-specific module includes membership matrices $\mathbf{H}_c$ ($c = 1, 2, \cdots, C$ for $C$ conditions) and expression levels matrices $\mathbf{W}_{b1}$ ($b = 1, 2, \cdots, B$ for $B$ batches). The unshared-bio factors are encoded in both $\mathbf{W}_{b1}$ and $\mathbf{H}_c$. We calculated the dot product of $\mathbf{W}_{b1}$ and $\mathbf{H}_c$, and obtained cell-by-gene matrices $\mathbf{X}_b^{\text{cond}}$ that only encoded the condition-related information. We treated $\mathbf{X}_b^{\text{cond}}$ as the unshared-bio factors of scINSIGHT.

When using scINSIGHT to detect CKGs, we set $K_j = 1$ such that the membership matrices $\mathbf{H}_c$ shrink into 1D vectors with length equal to the number of genes. We then treated $\mathbf{H}_c$s as the importance score of genes under condition $c$. The genes that have important scores varying the most across conditions should be the CKGs. We calculate the variance of gene $i$ under different conditions following:

$$\text{Var}(i) = \frac{1}{C}\sum_{c=1}^{C}\left(\mathbf{H}_c[i] - \frac{1}{C}\sum_{c=1}^{C}\mathbf{H}_c[i]\right)^2 \qquad (9)$$

We then normalize the variances of genes to the range of 0 and 1, which are used as the scoring of CKGs from scINSIGHT.

**Running scGen and scPreGEN.** We ran scGen following the steps and parameter setting in its online tutorial (https://scgen.readthedocs.io/en/stable/tutorials/scgen_perturbation_prediction.html). We ran scPreGAN following the parameter setting in its online repository (https://github.com/XiajieWei/scPreGAN-reproducibility).

In perturbation prediction tests, we trained scGen and scPreGAN on the datasets that are only normalized with library size. As the test requires the model to generate the count matrix as close as the gold-standard count matrix, and additional preprocessing steps introduce unnecessary bias. Both scGen and scPreGAN are designed for the perturbation prediction task where only one condition type is

involved. To predict the condition effect on datasets where two condition types are involved (simulated and COVID-19 datasets), we ran both methods on a subset of count matrices where one condition type is fixed and another condition varies, and the methods are trained to predict the condition effect of the changed condition. To further predict the joint effect of two condition types (e.g., disease severity and *age* in COVID-19 dataset), two cascade models are needed for both methods, where the first model learns one condition effect and the second model learns the other condition effect. For the test datasets, the detailed settings of training data used in both methods are described in Supplementary Tables 2, 3, 6–8.

### Evaluation metrics and gold-standard data

We evaluate the disentanglement of biological factors and technical batch effect using the ARI (adjusted rand index) and ASW-batch (batch-mixing average silhouette width) scores[38]. The ARI score measures the matching of latent space cluster result and ground truth cell type label. An ARI score ranges from 0 to 1, and a higher score means better separation of cell types in the latent space. The ASW-batch scores measure how well the cells of the same cell type are aligned among batches in the latent space[38]. The scores range between 0 and 1, and a higher score means a better alignment of batches and removal of batch effect.

To evaluate the CKGs detection accuracy, we use AUPRC (area under the precision-recall curve) score, Early Precision score (Eprec)[39] and Pearson correlation score. The scores are calculated using the ground truth CKGs and inferred CKGs, and a higher score means better detection accuracy.

When evaluating perturbation prediction, we use the denoised count matrix as gold-standard for scDisInFact. This is because the raw count matrix has technical noise, while the scDisInFact prediction is already denoised thanks to the use of the negative binomial distribution in the likelihood function. Directly comparing scDisInFact prediction with the raw counts would also induce the error caused by technical noise. We instead pass the counts through the scDisInFact model without updating any latent factors to generate the denoised output, and compare the predicted counts with the denoised counts instead. We still compare the prediction result of scGen and scPreGAN with the raw count directly, since these methods did not model the technical noise in their prediction.

We evaluate the prediction accuracy of gene expression data using MSE (mean square error), Pearson correlation, and $R^2$ scores. We calculate the scores on count matrices that are normalized against library size. MSE measures the difference between true and predicted count matrices on all input genes, where smaller values are better. Pearson correlation also measures the alignment of true and predicted count matrices. It ranges between −1 and 1, and a higher Pearson correlation means a better prediction result. $R^2$ measures the coefficient of determination. A higher $R^2$ score means a better matching between the predicted and true counts, and the maximum $R^2$ score is 1. In the cases where there is a one-to-one correspondence between the gold-standard counts and the predicted counts, MSE, Pearson correlation, and $R^2$ scores can be directly measured between cells. In the cases where the gold-standard counts and the predicted counts are not from the same cell, there is no direct way to calculate MSE, Pearson correlation, and $R^2$ for each cell. We instead calculate the score in a cell-type-specific manner. We calculate the centroid of each cell type using the mean gene expression data, and then measure the MSE, Pearson correlation, and $R^2$ between the centroid gene expression value of predicted and gold-standard counts for each cell type. In our simulation test and tests on GBM and COVID-19 datasets, we used cell-type-level scores as there is no one-one-correspondence between the gold-standard and predicted counts. In our ablation and hyper-parameter tests, we used cell-level scores since we have the cell-level ground truth.

### Reporting summary

Further information on research design is available in the Nature Portfolio Reporting Summary linked to this article.

## Data availability

All datasets used in this paper are previously published and freely available. The glioblastoma (GBM) dataset is available at Gene Expression Omnibus under accession number GSE148842. The COVID-19 dataset is downloaded from the website with the link: https://atlas.fredhutch.org/fredhutch/covid/. The original data are also available at Gene Expression Omnibus under accession numbers GSE155673, GSE149689, and GSE150728. All testing datasets and source data are available through Zenodo[41]. Source data are provided with this paper.

## Code availability

The code of scDisInFact is available on GitHub with the link: https://github.com/ZhangLabGT/scDisInFact. The package version used for the analyses in the paper has been assigned a citable DOI through Zenodo[42].

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

## Acknowledgements

This work was supported in part by the US National Science Foundation DBI-2019771, DBI-2145736 and National Institutes of Health grant R35GM143070 (Z.Z., X.Zhang). The authors would like to thank Dr. Yu Li for helpful discussions on this project. The human icons in Fig. 1 were created with BioRender.com and the neural networks in Figs. 1 and S1 were created with NN-SVG[40].

## Author contributions

Z.Z. and X.Zhang conceived the idea of scMoMaT. Z.Z. and X.Zhao implemented the scMoMaT algorithm. Z.Z. and M.B. carried out the evaluation and data analysis. P.Q. helped with the evaluation and data analysis. Z.Z. and X.Zhang wrote the paper. X.Zhang supervised the work.

## Competing interests

The authors declare no competing interests.
