## [Peer Review File · Nature Communications]

scDisInFact: disentangled learning for integration and prediction of multi-batch multi-condition single-cell RNA-sequencing dataREVIEWER COMMENTS

Reviewer #1 (Remarks to the Author):

In this paper, the authors developed a VAE based model scDisInFact to decouple condition and batch effects of single cell RNA-seq datasets. The model learns shared latent spaces from multiple conditions and condition specific latent spaces as well as batch related ones for removing batches and identifying condition specific genes. The pre-trained models also enable predicting outcomes of gene perturbations.

Overall, scDisInFact is novel, timely and sophisticated for single cell data processing & analysis, especially given that many single cell sequencing data emerge from multiple conditions and batches. It is appreciated that scDisInFact is able to carry out multiple essential jobs simultaneously for single-cell analysis, batch removal, condition-specific gene identification, and prediction. The paper was well written with the clear technic statement on the improvements of scDisInFact over the state of the arts. The authors demonstrated their outperformance using simulated data and real datasets (brain tumor, covid). I just have a few comments or suggestions:

- Can different condition types share or co-vary conditions?
- Are the shared latent spaces shared across all conditions or some? The model seems not to consider partially shared latent spaces.
- Cell types are mainly defined by marker genes. Is it possible that scDisInFact reveals novel or rare cell clusters (or types)?
- The CKG scores were calculated by AUPRC. Is there a particular reason like due to imbalance? The authors may provide other metrics for justification.
- It is great that the paper applied to multiple covid datasets. However, it is unclear whether the diagnosis of disease conditions is consistent across datasets. For example, were moderate or severe symptoms defined robustly or arbitrarily in each dataset?
- Aligning condition latent spaces (shared or unshared) by MMD across batches may be biased by always mapping to 1st batch. Is there any way to define such 1st batch? Will pairwise MMDs of conditions help?
- The authors may need to justify the necessity of each loss function, such as discriminator, and elaborate on how each loss contributes to the performance. Ablation tests may help.
- Providing running times and comparison would be appreciated. Some guidelines for biologist users can also help.

Reviewer #2 (Remarks to the Author):

Zhang et al developed a new computational method, scDisInFact, to study single-cell RNA-seq data by providing a simultaneous solution of batch effect removal, condition-dependent gene identification, and simulation of unobserved batch/condition-specific gene expression profiles based on other observed data. scDisInFact enables batch effect removal for multiple batch data, which is an important and challenging task. The perturbation analysis for the simulation of unobserved batch/condition-specific gene expression profiles is very ambitious. This is a very difficult task. Based on the authors' experiment, looks like this task can be handled at a certain level by scDisInFact. The manuscript is well-written and organized. The mathematical formulation and computational considerations are well-presented and easy to follow. Overall, a new capability that addresses three important tasks for scRNA-seq data analysis has been developed here. A few concerns need to be addressed:

1. Mathematical formulation:

I understand for simplicity the authors only illustrate the computation of one condition in METHODS. However, it is still unclear how the KL distances $KL(Q(Z_u | b, x) || P(Z_u, b))$ and $KL(Q(Z_s | b, x) || P(Z_s, b))$ could ensure the computation of shared-bio/unshared-bio factors. Based on the description, looks like the batch ID and condition were not directly input in scDisInFact. Instead, $P(Z_s, b)$ and $P(Z_u, b)$ are used. It is unclear how these two distributions are computed without using the labels. Also, when there are multiple conditions, will one VAE be trained for each condition? Based on the current formulation, looks like the condition label were ignored by the method.

Independence of Z_u and Z_s is a very strong assumption considering the interactive effect of batch, condition, and cell type-specific variations. The authors are expected to justify this assumption.

The perturbation analysis may oversimplify the problem, especially considering interactive effects cannot be handled. The major concern about this analysis is the identifiability of the true expression of one condition/batch by the generation approach trained without observing it. The method may work if the batch effect is consistent with all cells/conditions and the batch/condition effect is truly additive in the embedding space. I personally felt this is a very strong assumption and needs to be justified. I recommend the authors add more discussion about the mathematical assumptions of this analysis.

2. The methods have been well validated on synthetic data and real-world data. Ablation analysis is expected to test the necessity of the independence of Z_u and Z_s or remove one of them.

3. To ensure a robust analysis for each task, how many batches/conditions could be handled? How many samples need to be collected for each batch/condition and what proportion of the condition/batch can be missing and rescued by the perturbation analysis? And how many cells are needed for each condition/batch? Robustness analysis is needed.

4. If possible, I recommend the authors provide some theoretical discussions about the necessary assumptions to ensure the identifiability of the problems. This may help users to understand what assumptions were made when using this method.

Reviewer #3 (Remarks to the Author):

This work proposes a Variational Autoencoder-based method that aims to disentangle the batch, condition, and common effects of multi-batch and multi-condition single-cell RNAseq data in separate latent representations. The method was evaluated on simulated and real datasets to perform tasks including batch effect removal, condition-associated key gene detection, and perturbation prediction. Results indicate its superior performance to baseline methods including scINSIGHAT and scGen/scPreGAN.

In general, this disentanglement task is challenging, especially when batch and condition are one-on-one aligned such that none of the conditions are shared in more than one batch. Nonetheless, the approach proposed in this study sheds light on addressing the problem when conditions are shared across multiple batches.

I have a generally positive view of this work, yet there are still some concerns regarding the technical details.

1. The deep learning framework is trained with multi-tasking and in each iteration there are 3 consecutive steps for parameter update, which makes the training procedure complicated and challenging. Therefore, a detailed report showing the learning behavior of the model would be a helpful reference for replication or applying the method to new datasets. Key items include convergence curves of overall and separate losses, learning rate configuration, training and inference efficiency, etc.

2. The multi-tasking loss weights vary a lot, some are very small (e.g., $1e-5$, $1e-4$) compared to others (1.0). A sensitivity analysis of the hyper-parameters and/or an ablation study of some of the auxiliary losses might be helpful to understand the importance of different losses.

3. The authors mentioned in Section 4.5 that “Users can manually tune the hyper-parameters around the recommended setting instead of conducting a comprehensive grid search, and it should provide a

reasonable result on most of the dataset.”. Following the above comments, would the authors give more detailed instructions on how to diagnose model training and judge whether on a new dataset the model has succeeded or failed?

4. In 2.2.2 when comparing with scINSIGHT, training two models might result in unaligned latent representations, a fairer way might be to create a single new condition using Cartesian product of the original two conditions.

Other minor comments:

In Fig. 2d the middle sub-figure, the performance of scDisinfect looks worse than scINSIGHT with batch correction score, would the authors elaborate more on this result?

The text in Section 4.5 describing model architecture might be better visualized in a compact diagram.

Responses to Reviewers' Comments

We thank the reviewers for their time to review our manuscript and their valuable comments.

Please find below the reviewers' comments (in black) and our pointwise responses (in blue). The changes in the manuscript are marked in red.

Ziqi Zhang, Xinye Zhao, Mehak Bindra, Peng Qiu, Xiuwei Zhang

Reviewer #1

In this paper, the authors developed a VAE based model scDisInFact to decouple condition and batch effects of single cell RNA-seq datasets. The model learns shared latent spaces from multiple conditions and condition specific latent spaces as well as batch related ones for removing batches and identifying condition specific genes. The pre-trained models also enable predicting outcomes of gene perturbations.

Overall, scDisInFact is novel, timely and sophisticated for single cell data processing & analysis, especially given that many single cell sequencing data emerge from multiple conditions and batches. It is appreciated that scDisInFact is able to carry out multiple essential jobs simultaneously for single-cell analysis, batch removal, condition-specific gene identification, and prediction. The paper was well written with the clear technical statement on the improvements of scDisInFact over the state of the arts. The authors demonstrated their outperformance using simulated data and real datasets (brain tumor, covid). I just have a few comments or suggestions:

1. Can different condition types share or co-vary conditions?

This is a great question. In the scDisInFact model, we assume that different condition types have independent effects on the data. The assumption is to ensure the disentanglement of latent factors and the accuracy of perturbation prediction. To predict the gene expression data under a condition of a certain condition type, scDisInFact updates the corresponding unshared-bio factors (Page 19, Lines 530-546, Fig. S1b in the manuscript). If the condition type is correlated to some other condition types, then unshared-bio factors corresponding to all correlated condition types should also be considered, which makes it hard for perturbation prediction. When there exist correlated condition types, we suggest the users combine them into one condition type using the Cartesian product of the original condition types instead of dealing with each condition type separately. For example, if condition type 1 (stim1, ctrl1) and condition type 2 (stim2, ctrl2) are correlated, the users can create a combined condition type (stim1_stim2, stim1_ctrl2, ctrl1_stim2, ctrl1_ctrl2).

We also added discussion on this assumption in the Discussion section of the manuscript (Page 15, Lines 431-448).

2. Are the shared latent spaces shared across all conditions or some? The model seems not to consider partially shared latent spaces.

We thank the reviewer for this question, which gives us an opportunity to show the performance of scDisInFact in the case of partially shared latent spaces. In scDisInFact, the shared latent space across count matrices are named shared-bio factors. In this response, first, we confirm that the shared-bio factors are designed to capture information shared across all input matrices, and in our original submission, we used datasets where all data matrices have the same set of cell types. Second, we would like to add that scDisInFact can also work with the case where data matrices do not share the same set of cell types to a certain extent. This is because: although the MMD loss is applied to the cells' latent embedding to align the distribution of the cells' latent embedding across all conditions and batches, the final embedding is determined also by other factors (eg. reconstruction loss) in addition to MMD loss. By controlling the weight of MMD loss relative to other loss terms, scDisInFact can learn latent embedding that is partially aligned across conditions or batches.

Next, we show results of scDisInFact where data from certain conditions or batches have one or two cell types missing compared to the majority of data matrices. We conducted tests on the following simulated datasets:

1. We first test the case where the latent embedding is partially aligned (mismatched) across batches. We used the same simulated datasets in the manuscript (described in Sec. 2.2), and removed one cell type (cell type 16) or two cell types (cell types 10 and 16) in data matrices of one data batch (batch 0, meaning half of the input data matrices have these cell types removed) to create datasets with partially aligned latent embedding across batches. We ran scDisInFact on the processed datasets with the default hyper-parameter setting in the simulation test, and evaluated the shared-bio factors in aligning the latent embedding of the partially aligned datasets using ARI and ASW-batch scores. Same as the simulation test in the main manuscript, ARI score measures the separation of cell types, whereas ASW-batch measures the alignment of latent embedding.

Figure R1. The ARI scores and ASW-batch scores of scDisInFact when (1) all cell types are matched across batches (All CT), (2) one cell type is removed from one batch (1 CT rem), and (3) two cell types are removed from one batch (2 CT rem).

In the result (Fig. R1), both ARI and ASW-batch scores decrease with the increase of the degree of mismatch across batches, which was expected. However, scDisInFact can still maintain a decent performance when there exists mismatch across batches (ARI ≥ 0.7 , ASW-batch > 0.95).

2. We now test the performance of scDisInFact when there exists cell type mismatch across conditions. We remove all cells of cell type 16 or cell type 10 and 16 from the simulated data under healthy conditions to simulate the dataset with different degrees of mismatch across conditions (again, half of the input matrices have these cell types removed). We show the ARI and ASW-batch scores of scDisInFact on the processed datasets with the default hyper-parameter below:

Figure R2. The ARI scores and ASW-batch scores of scDisInFact when (1) all cell types are matched across conditions (All CT), (2) one cell type is removed from healthy condition (1 CT rem), and (3) two cell types are removed from healthy condition (2 CT rem).

From the result (Fig. R2), we again observe that both ARI and ASW-batch scores decrease with the increase of degree of mismatch across conditions. Overall, scDisInFact can still maintain a good performance when there exists mismatch across conditions (ARI ≥ 0.7 , ASW-batch > 0.95).

We also include these results and corresponding discussions in the manuscript (Sec. 2.2.2, Page 7 Lines 168-180, Supplementary Note 1, Figs. S3).

3. Cell types are mainly defined by marker genes. Is it possible that scDisInFact reveals novel or rare cell clusters (or types)?

This is a great question. We did not demonstrate that scDisInFact can help with detecting rare cell types in our original submission. Here we show scDisInFact can: (1) generate a cleaner cell type pattern using shared-bio factors which are removed of batch effects and condition effects; (2) increase the total number of cells from rare cell types by integrating multiple data matrices.

Based on the simulated datasets (dataset structure follows Fig. 2a) used in the manuscript, we created the following challenging scenario, where a cell type is not only rare in some data matrices, but also missing in the rest of matrices. We set cell type 4 to be the rare cell type, and then:

1. Randomly select 20% of cells from cell type 4 and remove the rest of the cells of cell type 4.
2. Removed all the cells of cell type 4 from data matrices under the “stim” condition.

We visualize the data matrices before applying scDisInFact using UMAP (Fig. R3 below). One can observe that it is challenging to identify the rare cell type using each count matrix alone, not only because of the small number of cells, but also because the latent embedding of cells is confounded by batch and condition effect, which makes the cell type pattern noisy. :

Figure R3. UMAP visualization of the gene expression data, where cells are visualized separately according to the conditions and batches. Cells colored by (a) ground truth cell type annotation, and (b) the rare cell type annotation (rare cell type: cell type 4).

We then ran scDisInFact on the dataset and visualized the shared-bio factors using UMAP:

Figure R4. UMAP visualization of the shared-bio factors learned by scDisInFact, where cells are visualized separately according to the conditions and batches. Cells are colored by (a) ground truth cell type annotation, and (b) the rare cell type annotation (rare cell type: cell type 4, within red circle).

Figure R5. UMAP visualization of the unshared-bio factors of cells from all input matrices learned by scDisInFact. The cells are visualized jointly across batches and conditions, and are colored by rare cell type annotation (rare cell type: cell type 4)

From the UMAP visualization of the shared-bio factors, we can see that cells of cell type 4 are separated from the other cells (Figs. R4b, R5), which makes it straightforward to detect this cell type.

We also included these results in the manuscript (Sec. 2.2.2, Page 7 Lines 168-180, Supplementary Note 2, Figs. S4,S5a).

4. The CKG scores were calculated by AUPRC. Is there a particular reason like due to imbalance? The authors may provide other metrics for justification.

Yes, we use AUPRC because the number of positive genes is far less than the number of negative genes in the test (positive genes: 20, 50, 100, total genes: 500), which creates an imbalanced scenario. According to the reviewer's suggestions, we also include the early precision score (Pratapa et al. 2020) and Pearson correlation as additional evaluation metrics. Both metrics show that scDisInFact outperforms baseline methods, consistent with the AUPRC results shown in our original submission:

Figure R6. Early precision scores of scDisInFact and baseline methods on simulated datasets with different perturbation parameters.

Figure R7. Pearson correlation scores of scDisInFact and baseline methods on simulated datasets with different perturbation parameters.

We included these results and added corresponding discussions in the manuscript (Page 8, Lines 197-200, Fig. S5b).

5. It is great that the paper applied to multiple covid datasets. However, it is unclear whether the diagnosis of disease conditions is consistent across datasets. For example, were moderate or severe symptoms defined robustly or arbitrarily in each dataset?

We agree that the classification standard of condition should be made consistent when dealing with datasets from different sources. We also noticed that the classification of disease severity from different papers were hard to be compared since the evaluation metrics vary across different experimental groups. Here instead of directly using the severity classification result from the original datasets, we adopted standardized severity classification proposed in the summary paper (Tian et al. 2022). In the summary paper, the authors gathered the COVID-19 datasets from different sources, including the three datasets that we use, and standardized the evaluation metrics across different datasets to create a comparable disease severity condition

(healthy=healthy control, moderate=hospitalized but no ICU, severe=hospitalized and ICU). We use their classification as the disease severity label in our test.

6. Aligning condition latent spaces (shared or unshared) by MMD across batches may be biased by always mapping to 1st batch. Is there any way to define such 1st batch? Will pairwise MMDs of conditions help?

We agree that using the term “1st batch” is an arbitrary choice and can be confusing. To better define the reference batch in the MMD loss, scDisInFact now selects the largest batch (with the largest number of cells) as the reference batch for MMD calculation instead of the “1st batch” in the original manuscript. We updated all our existing test results accordingly. The model still maintains a similar performance in all the tests.

We also considered using pairwise MMDs loss, but after careful consideration, we decided to use the single-reference MMD loss because this is computationally much more efficient than calculating pairwise MMD loss. Suppose there are n batches, in the pairwise manner, we need to compute MMD loss $O(n^2)$ times, whereas in the single-reference manner, we only need to compute MMD loss $O(n)$ times.

We have updated the manuscript to describe the new way to choose the reference batch when calculating the MMD loss (Page 17, Lines 483-505).

7. The authors may need to justify the necessity of each loss function, such as discriminator, and elaborate on how each loss contributes to the performance. Ablation tests may help.

We thank the reviewer for this valuable comment. This comment is related to some comments from other reviewers (Q7 of reviewer #1, Q1&2 of reviewer #2, Q2 of reviewer #3), and we address these comments together.

After conducting careful ablation tests on the loss terms of our original model, we learned that indeed there is redundancy in our original model. Therefore, we removed the redundant parts of the model while maintaining the performance of the model. The updates on the model are summarized below:

1. We removed the discriminator and the total correlation loss. Our ablation test does not support that the total correlation loss improves the overall performance of the model, and the independence assumption between shared-bio factors and unshared-bio factors enforced by the total correlation loss is too strong.

2. We removed the contrastive loss. Given the existing classifier in the model, the additional contrastive loss does not further improve the overall performance, especially the perturbation prediction accuracy. The ablation test did not show the advantage of including the contrastive loss.

3. We changed the prior distribution of unshared-bio factors from standard normal distributions into Gaussian Mixture distributions (each condition label corresponds to one Gaussian, and the means and variances of Gaussians are learned from data). The standard normal prior distribution does not match the true prior of the unshared-bio factor, where factors of different conditions should be distinguished.

The details of the updated model are included in the manuscript (Pages 16-17 Lines 468-505). After updating the model, we re-ran all the tests in the manuscript, and replaced the results and the corresponding figures. The overall performance of the model remains the same, but the complexity of the model is reduced.

We conducted the ablation test on the updated model to justify the necessity of the remaining loss function. There are totally 5 loss terms in the updated model: the reconstruction loss, the KL divergence to the prior distribution, the MMD loss on the latent factors, the classification loss on the unshared-bio factor, and the group-lasso loss on the unshared encoders (Eq. 6 in the manuscript).

Since the reconstruction loss and the KL divergence loss (which together make the ELBO loss) are essential terms for a VAE model, we conducted the ablation test on the remaining three losses: MMD loss, classification loss, and group-lasso loss. We conducted the ablation test using the same simulated datasets used in the manuscript (Sec. 2.2, totally 9 datasets). In the model training procedure, we held out the counts under the condition $\langle ctrl, severe \rangle$, of batches 1&2, and trained the model using the remaining count matrices. We describe the ablation tests of each loss term below:

1. *MMD loss*. In scDisInFact, the use of MMD loss serves three purposes:

- a. Removing batch effects from shared-bio and unshared-bio factors, which means that the shared-bio/unshared-bio factors of different batches should follow the same distribution.
- b. Removing condition effects from shared-bio factors, which means that the shared-bio factors of different conditions should follow the same distribution.
- c. Disentangling unshared-bio factors of different condition types. That is to make sure that the unshared-bio factors of a certain condition type do not encode the condition effect of other condition types.

We use ASW scores (described in Sec. 4.9 in the manuscript) to measure the effectiveness of the MMD loss in terms of the three purposes above. Fig. R8a-c shows the ASW scores to measure respectively: (a) the alignment of shared-bio and unshared-bio factors across different batches (Fig. R8a); (b) the alignment of shared-bio factors across different conditions (Fig. R8b). (c) the alignment of unshared-bio factors across different conditions of other condition types (except for the condition type it encodes, Fig. R8c). These results show overall better ASW scores when using MMD loss (reg:0.0001) than not using MMD loss (reg:0), which means that the use of MMD loss helps achieve the three purposes above.

Figure R8. Ablation test of MMD loss (no MMD, reg: 0; MMD, reg: 0.0001) **a.** Alignment of shared-bio and unshared-bio factors across batches. **b.** Alignment of shared-bio factors across different conditions. **c.** alignment of unshared-bio factors across different conditions.

The use of MMD loss can further affect the accuracy of perturbation prediction. We test the accuracy of perturbation prediction with and without using MMD loss. We use count matrix of different conditions as input and predict the corresponding count matrix in the held-out condition (ctrl & severe & batch 0):

- Input stim & severe & batch 0, predict “Treatment”;
- Input ctrl & healthy & batch 0, predict “Severity”;
- Input stim & healthy & batch 0, predict “Treatment & Severity”;

Since we have the ground truth count matrix, we directly measure the MSE between the predicted counts and the ground truth counts for each cell, and report the average MSE. The result shows that the use of MMD loss improves the perturbation prediction accuracy (Fig. R9).

Figure R9. Perturbation prediction accuracy with MMD loss (reg: 0.0001) and without MMD loss (reg: 0).

2. *The classification loss.* The classification loss separates unshared-bio factors corresponding to different conditions, which is critical for the perturbation prediction accuracy of the model. We

trained the model with and without classification loss (regularization weights 0 and 1), and calculated the perturbation prediction accuracy using MSE. The results show that the model with classification loss (reg: 1) shows lower MSE compared to the model without classification loss (Fig. R10).

Figure R10. Perturbation prediction accuracy (measured with MSE) of the model with and without classification loss (reg: 0 & reg: 1)

3. *The group-lasso loss.* The group-lasso loss helps the model to extract the most important CKGs. We trained the model with and without group-lasso loss, and calculated the CKGs detection using AUPRC (the same metric in the manuscript). As expected, the use of group-lasso loss (reg: 1) significantly improves the CKGs detection accuracy of the model (Fig. R11).

Figure R11. Perturbation prediction accuracy (measured with MSE) of the model with and without group-lasso loss (reg: 0 & reg: 1)

We have included the ablation test results (Figs. S12,S13) and corresponding discussions in the manuscript (Sec. 2.5.1, Supplementary Note 3).

8. Providing running times and comparisons would be appreciated. Some guidelines for biologist users can also help.

We thank the reviewer for the suggestion. We ran scDisInFact and baseline models using datasets of different sizes, and compared their running time (results added to Fig. S15 and discussed in Sec. 2.6, Page 14, Lines 401-405). scDisInFact scales well with the increase in data size, which is similar to the other deep-learning-based model. These models all use mini-batch stochastic gradient descent, which makes it possible for them to deal with large

datasets through the use of GPUs. scINSIGHT, on the contrary, does not scale well on large datasets.

Figure R12. Running time of scDisInFact and baseline methods on datasets of different sizes. X-axis shows the total number of cells within the dataset.

According to the reviewer's suggestions, we incorporate more running details on our GitHub page (<https://github.com/ZhangLabGT/scDisInFact>) to help the biologists with running the method, which includes the details of package installation, introduction to basic running commands, and a demo script for reference. In addition, we also provide guidance on hyper-parameter tuning of the model in the manuscript to better assist the biologist when they want to adjust the results of the method (Pages 19-20, Lines 559-574).

Reviewer #2

Zhang et al developed a new computational method, scDisInFact, to study single-cell RNA-seq data by providing a simultaneous solution of batch effect removal, condition-dependent gene identification, and simulation of unobserved batch/condition-specific gene expression profiles based on other observed data. scDisInFact enables batch effect removal for multiple batch data, which is an important and challenging task. The perturbation analysis for the simulation of unobserved batch/condition-specific gene expression profiles is very ambitious. This is a very difficult task. Based on the authors' experiment, looks like this task can be handled at a certain level by scDisInFact. The manuscript is well-written and organized. The mathematical formulation and computational considerations are well-presented and easy to follow. Overall, a new capability that addresses three important tasks for scRNA-seq data analysis has been developed here. A few concerns need to be addressed:

1. Mathematical formulation:

I understand for simplicity the authors only illustrate the computation of one condition in METHODS. However, it is still unclear how the KL distances $KL(Q(Z_u|b,x)||P(Z_u,b))$ and $KL(Q(Z_s|b,x)||P(Z_s,b))$ could ensure the computation of shared-bio/unshared-bio factors. Based on the description, looks like the batch ID and condition were not directly input in scDisInFact. Instead, $P(Z_s,b)$ and $P(Z_u,b)$ are used. It is unclear how these two distributions

are computed without using the labels. Also, when there are multiple conditions, will one VAE be trained for each condition? Based on the current formulation, looks like the condition label were ignored by the method.

We thank the reviewer for raising these questions. Below we summarize the comment into 4 questions, and provide the answer to each question separately:

Q1.1: Where are the condition labels used in the model and how do the labels help the model to learn \mathbf{z}_u and \mathbf{z}_s ?

A1.1: Condition labels are part of the input to `scDisInFact`, but it is not directly fed into the encoder similar to the gene expression data or batch ID.

In the original version of `scDisInFact` (prior to the revision), the condition labels were used as the ground truth for the classifier. The classifier predicts the condition labels from the unshared-bio factors \mathbf{z}_u . When predicting the condition labels from \mathbf{z}_u , we update both the unshared encoder (which means also updating \mathbf{z}_u) and the classifier to improve the condition label prediction accuracy. After training, the classifier is capable of predicting the condition labels from \mathbf{z}_u , and \mathbf{z}_u at the same time also preserves the biological information for condition label prediction. As for \mathbf{z}_s , the condition label is only used in the MMD loss of \mathbf{z}_s to make sure that \mathbf{z}_s is removed from the condition effect.

In the updated version of `scDisInFact`, the condition labels are also used in the KL divergence term of unshared-bio factors ($KL(Q(\mathbf{z}_u|\mathbf{x})||P(\mathbf{z}_u|c))$) in addition to the classifier ground truth. This is because we changed the prior distribution of unshared-bio factors from standard normal distributions into Gaussian Mixture distributions (where each condition label corresponds to one Gaussian, and the means and variances of Gaussians are learned from data). Detailed information on the formulation is in our response to the reviewer's Comment #2 and the updated Methods in the main manuscript (Pages 16-17, Lines 468-505).

Q1.2: How are the KL divergence terms $KL(Q(\mathbf{z}_u|b,\mathbf{x})||P(\mathbf{z}_u,b))$ and $KL(Q(\mathbf{z}_s|b,\mathbf{x})||P(\mathbf{z}_s,b))$ used to ensure the computation of shared-bio factors and unshared-bio factors?

A1.2: The two KL divergence terms ensure that the learned posterior distribution $Q(\mathbf{z}_s|b,\mathbf{x})$ and $Q(\mathbf{z}_u|b,\mathbf{x})$ are close to their predefined prior distribution. The terms are parts of the ELBO loss of VAE model. However, the posterior distributions of shared-bio and unshared-bio factors ($Q(\mathbf{z}_s|b,\mathbf{x})$ and $Q(\mathbf{z}_u|b,\mathbf{x})$, which are used to sample the shared-bio and unshared-bio factors) are primarily learned through the reconstruction loss and classification loss. These two losses make sure that $Q(\mathbf{z}_s|b,\mathbf{x})$ and $Q(\mathbf{z}_u|b,\mathbf{x})$ encode the major biological information within the data.

Q1.3: How is the batch ID used in the model?

A1.3: The batch ID is used as the input of the encoder together with the gene expression data. The shared encoder takes as input the concatenated gene expression data and batch ID

(Figure 1 in the main manuscript). The shared encoder models the posterior distribution $Q(\mathbf{z}_s|\mathbf{x},b)$ (input is gene expression \mathbf{x} and batch ID b , output is \mathbf{z}_s).

The unshared encoder models the $Q(\mathbf{z}_u|\mathbf{x})$, and batch ID b is not input into the unshared encoders. We noticed that we used the notation $Q(\mathbf{z}_u|\mathbf{x},b)$ which can be confusing, and in our updated manuscript, we replaced $Q(\mathbf{z}_u|\mathbf{x},b)$ by $Q(\mathbf{z}_u|\mathbf{x})$ (Page 16, Eq. 2).

In the manuscript, the prior distribution was written as $P(\mathbf{z}_u|b)$ and $P(\mathbf{z}_s|b)$ as a general form derived from the evidence lower bound, but in fact, the batch ID was not used to define the distribution. For example, we have $P(\mathbf{z}_s|b)=P(\mathbf{z}_s)$ (detailed explanation in Page 16, Lines 468-475, Eq. 6).

Q1.4: When there are multiple conditions, will one VAE be trained for each condition?

A1.4: The VAE model used in scDisInFact is not a standard VAE model (with one encoder and one decoder), but a customized one with multiple encoders and one decoder. The encoders include one shared encoder and several unshared encoders depending on the number of condition types, where each condition type corresponds to one unshared encoder (Fig. 1). The number of unshared encoders is decided by the number of condition types. In the GBM dataset, we only have one condition type (Fig. S7a, panobinostat treatment or not), then we only use one unshared encoder. In the COVID-19 dataset, we have two condition types (Fig. 5a, disease severity and age), then there are two unshared encoders. The decoder then takes as input both the shared-bio factor learned by the shared encoder and several unshared-bio factors learned by the unshared encoders, and reconstructs the input gene expression data. The training of all encoders and the decoder is conducted together.

We updated the method description correspondingly to make it easier to understand how scDisInFact works (Sec. 4.1, Pages 16-17).

Independence of Z_u and Z_s is a very strong assumption considering the interactive effect of batch, condition, and cell type-specific variations. The authors are expected to justify this assumption.

This is an excellent question. Indeed the independence of \mathbf{z}_u and \mathbf{z}_s is not realistic in the cases when perturbations trigger cell-type-specific responses. In such cases, the independence of \mathbf{z}_u and \mathbf{z}_s should not be enforced. Based on this consideration and the ablation test results, we have removed the discriminator and the total correlation loss that enforces the independence of \mathbf{z}_u and \mathbf{z}_s in the model in this revision (please see our responses to the reviewer's Comment #2 for details of the model updates). We updated the test results in the manuscripts with the new model, and we were able to maintain the same level of performance. In addition, we also added analysis on all the model assumptions in the Discussion section (Page 15, Lines 431-448).

The perturbation analysis may oversimplify the problem, especially considering interactive effects cannot be handled. The major concern about this analysis is the identifiability of the true expression of one condition/batch by the generation approach trained without observing it. The

method may work if the batch effect is consistent with all cells/conditions and the batch/condition effect is truly additive in the embedding space. I personally felt this is a very strong assumption and needs to be justified. I recommend the authors add more discussion about the mathematical assumptions of this analysis.

This is a great point. We acknowledge that we have made the two assumptions mentioned in this comment: (1) the batch effect is consistent across conditions, and (2) the batch/condition effects are additive in the embedding space.

Firstly, we assume that the condition effect and batch effect are two independent effects in the data. That is, $P(b|c)=P(b)$, where $P(b)$ is the distribution of batch effects, and c is the random variable representing condition effects. The assumption is made considering both the underlying mechanism that causes the batch/condition effects and the feasibility of data modeling. Batch effect is mainly caused by the technical confounders from experiments, and on the other hand, condition effect is caused by biological variance that exists in the experimental subjects. We understand that batch effect and condition effect originate from different sources does not justify that they are independent, and we can not exclude the cases where batch effects are related to conditions. However, the relationship between batch effects and condition effects, if any, is unknown to the current research community. Existing works either consider only batch effect or only condition effect (Lotfollahi, Wolf, and Theis 2019; Wei, Dong, and Wang 2022) and scDisInFact seeks to take one step further by considering both batch and condition effects. In order to do this, we have to make an assumption on the relationship between batch and condition effect, and we choose the independence assumption given the lack of knowledge on this relationship. This assumption allows us to predict the gene expression data under the unseen conditions with better performance than existing methods that consider only condition effects (which corresponds to a stronger assumption that the batch effects do not exist)..

Secondly, by concatenating the unshared-bio factor with the shared-bio factors in the latent space, the model implicitly assumes that the condition effect is additive to cell embedding on the latent space. This assumption has also been used in recent work for perturbation prediction (Lotfollahi, Wolf, and Theis 2019; Lotfollahi et al. 2023). In our opinion, the major reasons of this additive assumption being used in existing and our works are two folds: (1) the real mathematical relationship between condition effects are not known, thus a simple function (like the additive function) is a natural choice; (2) although the additive function applied to the latent embedding is linear and simple, the encoders which output the latent embedding has a few layers of nonlinear transformations which are expected to account for as much of the complexity in the data as possible through training.

We acknowledge that although we have improved over existing work, the current model is not ideal and does not model all complex relationships between different effects in the data. We anticipate that as further understanding on these relationships are gained, future method development can take this into account and lift some of the assumptions in current computational models. We include the full discussion on the model assumptions and their

corresponding limitations in modeling perturbation effects in the Discussion section of the manuscript too (Page 15, Lines 431-448).

2. The methods have been well validated on synthetic data and real-world data. Ablation analysis is expected to test the necessity of the independence of Z_u and Z_s or remove one of them.

We thank the reviewer for the suggestion. This comment is related to some comments from other reviewers (Q7 of reviewer #1, Q1&2 of reviewer #2, Q2 of reviewer #3), and we address these comments together.

After conducting careful ablation tests on the loss terms of our original model, we learned that indeed there is redundancy in our original model. Therefore, we removed the redundant parts of the model while maintaining the performance of the model, including those that enforce the independence of z_u and z_s , mentioned in this comment. The updates on the model are summarized below:

1. We removed the discriminator and the total correlation loss. Our ablation test does not support that the total correlation loss improves the overall performance of the model, and the independence assumption between shared-bio factors and unshared-bio factors enforced by the total correlation loss is too strong.

2. We removed the contrastive loss. Given the existing classifier in the model, the additional contrastive loss does not further improve the overall performance, especially the perturbation prediction accuracy. The ablation test did not show the advantage of including the contrastive loss.

3. We changed the prior distribution of unshared-bio factors from standard normal distributions into Gaussian Mixture distributions (each condition label corresponds to one Gaussian, and the means and variances of Gaussians are learned from data). The standard normal prior distribution does not match the true prior of the unshared-bio factor, where factors of different conditions should be distinguished.

The details of the updated model are included in the manuscript (Pages 16-17 Lines 468-505). After updating the model, we re-ran all the tests in the manuscript, and replaced the results and the corresponding figures. The overall performance of the model remains the same, but the complexity of the model is reduced.

We then conducted the ablation test on the updated model to justify the necessity of the remaining loss function. There are totally 5 loss terms in the updated model: the reconstruction loss, the KL divergence to the prior distribution, the MMD loss on the latent factors, the classification loss on the unshared-bio factor, and the group-lasso loss on the unshared encoders (Eq. 6 in the manuscript).

Since the reconstruction loss and the KL divergence loss (which together make the ELBO loss) are essential terms for a VAE model, we conducted the ablation test on the remaining three losses: MMD loss, classification loss, and group-lasso loss. We conducted the ablation test using the same simulated datasets used in the manuscript (Sec. 2.2, totally 9 datasets). In the model training procedure, we held out the counts under the condition $\langle ctrl, severe \rangle$, of batches 1&2, and trained the model using the remaining count matrices. We describe the ablation tests of each loss term below:

1. *MMD loss*. In scDisInFact, the use of MMD loss serves three purposes:

- a. Removing batch effects from shared-bio and unshared-bio factors, which means that the shared-bio/unshared-bio factors of different batches should follow the same distribution.
- b. Removing condition effects from shared-bio factors, which means that the shared-bio factors of different conditions should follow the same distribution.
- c. Disentangling unshared-bio factors of different condition types. That is to make sure that the unshared-bio factors of a certain condition type do not encode the condition effect of other condition types.

We use ASW scores (described in Sec. 4.9 in the manuscript) to measure the effectiveness of the MMD loss in terms of the three purposes above. Fig. R13a-c shows the ASW scores to measure respectively: (a) the alignment of shared-bio and unshared-bio factors across different batches (Fig. R13a); (b) the alignment of shared-bio factors across different conditions (Fig. R13b). (c) the alignment of unshared-bio factors across different conditions of other condition types (except for the condition type it encodes, Fig. R13c). These results show overall better ASW scores when using MMD loss (reg:0.0001) than not using MMD loss (reg:0), which means that the use of MMD loss helps achieve the three purposes above.

Figure R13. Ablation test of MMD loss (no MMD, reg: 0; MMD, reg: 0.0001) a. Alignment of shared-bio and unshared-bio factors across batches. b. Alignment of shared-bio factors across different conditions. c. alignment of unshared-bio factors across different conditions.

The use of MMD loss can further affect the accuracy of perturbation prediction. We test the accuracy of perturbation prediction with and without using MMD loss. We use count matrix of different conditions as input and predict the corresponding count matrix in the held-out condition (ctrl & severe & batch 0):

- a. Input stim & severe & batch 0, predict “Treatment”;
- b. Input ctrl & healthy & batch 0, predict “Severity”;
- c. Input stim & healthy & batch 0, predict “Treatment & Severity”;

Since we have the ground truth count matrix, we directly measure the MSE between the predicted counts and the ground truth counts for each cell, and report the average MSE. The result shows that the use of MMD loss improves the perturbation prediction accuracy (Fig. R14).

Figure R14. Perturbation prediction accuracy with MMD loss (reg: 0.0001) and without MMD loss (reg: 0).

2. The classification loss. The classification loss separates unshared-bio factors corresponding to different conditions, which is critical for the perturbation prediction accuracy of the model. We trained the model with and without classification loss (regularization weights 0 and 1), and calculated the perturbation prediction accuracy using MSE. The results show that the model with classification loss (reg: 1) shows lower MSE compared to the model without classification loss (Fig. R15).

Figure R15. Perturbation prediction accuracy (measured with MSE) of the model with and without classification loss (reg: 0 & reg: 1)

3. *The group-lasso loss.* The group-lasso loss helps the model to extract the most important CKGs. We trained the model with and without group-lasso loss, and calculated the CKGs detection using AUPRC (the same metric in the manuscript). As expected, the use of group-lasso loss (reg: 1) significantly improves the CKGs detection accuracy of the model (Fig. R16).

Figure R16. Perturbation prediction accuracy (measured with MSE) of the model with and without group-lasso loss (reg: 0 & reg: 1)

We have included the ablation test results (Figs. S12,S13) and corresponding discussions in the manuscript (Sec. 2.5.1, Supplementary Note 3).

3. To ensure a robust analysis for each task, how many batches/conditions could be handled? How many samples need to be collected for each batch/condition and what proportion of the condition/batch can be missing and rescued by the perturbation analysis? And how many cells are needed for each condition/batch? Robustness analysis is needed.

This comments consists of three questions and we answer each question separately below:

Q3.1: To ensure a robust analysis for each task, how many batches/conditions could be handled?

A3.1: scDisInFact does not have specific requirements on the numbers of batches or condition labels of the training data. Rather, scDisInFact performance can be affected if there are many missing data matrices (this is discussed in detail in our response to Q3.2 below). Here, suppose there are not too many missing matrices, we first discuss potential effects of the numbers of batches, condition labels, and condition types, and then test on simulated data of various scenarios.

Discussion: (i) More batches often corresponds to a larger total number of cells, which can be beneficial for training. The computational time for a large total number of cells is not a concern, as shown in Fig. S15. (ii) Similarly, for a given condition type (eg. age of patients), more condition labels (eg. 5 labels “20yo, 30yo, 40yo, 50yo, 60yo” compared to 3 labels “20yo, 40yo, 60yo”) can also provide more information for training. (iii) As for the number of condition types,

we expect that more condition types correspond to data of higher complexity and thus are more challenging for disentanglement of the condition types.

Tests and results:

(i) We first test how the model performs on datasets with different numbers of batches, we simulated two datasets of the following settings:

- a. 4 batches (denoted as b1, b2, ..., b4), 1 condition type, and 2 condition labels (denoted as c1 & c2), in total 10000 cells.
- b. 8 batches (denoted as b1, b2, ..., b8), 1 condition type, and 2 condition labels (denoted as c1 & c2), in total 20000 cells.

When training the model, we held out the data under condition c2 and batch b1 and trained the model on the remaining data. We then use the training data under condition c1 and batch b1 to predict the data under the held-out condition and batch (c2, b1). We measure the perturbation prediction accuracy using MSE and R2 scores. We then measure the condition-associated key genes (CKGs) detection accuracy using AUPRC. The results are as following:

Figure R17. The perturbation prediction accuracy (measured with MSE, R2) and the CKGs detection accuracy (AUPRC) of the model with different numbers of batches.

The result shows that scDisInFact has a better performance in perturbation prediction when there are more batches, which is expected as the dataset with more batches provides more training samples that help scDisInFact to learn the perturbation effect.

(ii) We then test the model performance on datasets with different numbers of condition labels. We simulated datasets with the following settings:

- a. 10000 cells, 2 batches (b1, b2), 1 condition type, and 3 condition labels (denoted as c1, c2, c3)
- b. 10000 cells, 2 batches (b1, b2), 1 condition type, and 4 condition labels (denoted as c1, c2, c3, c4)
- c. 10000 cells, 2 batches (b1, b2), 1 condition type, and 5 condition labels (denoted as c1, c2, c3, c4, c5)

We held out the data of condition c3 (3 condition labels)/c4 (4 condition labels)/c5 (5 condition labels) and batch b2. We then train the model on the remaining data. We use the

training data under condition c1 and batch b2 to predict the data under the held-out condition and batch. We measure the perturbation prediction accuracy using MSE and R2 scores. We then measure the condition-associated key genes (CKGs) detection accuracy using AUPRC. The results are as following:

Figure R19. The perturbation prediction accuracy (measured with MSE, R2) and the CKGs detection accuracy (AUPRC) of the model with different condition labels.

The final scores show that the model performs better when there are more condition labels (both CKGs detection and perturbation prediction). This can be due to the fact that more condition labels provide diverse training data which helps to learn the condition effect of a given condition type.

(iii) We finally test how scDisInFact performs on datasets with different numbers of condition types. We simulated datasets of the following settings:

- 10000 cells, 2 batches (b1, b2), 2 condition types, and 2 condition labels (2x2 combinations of condition labels, jointly denoted as c11, c12, c21, c22)
- 10000 cells, 2 batches (b1, b2), 3 condition types, and 2 condition labels (3x2 combinations of condition labels, jointly denoted as c111, c112, c121, c122, c211, c212, c221, c222)
- 10000 cells, 2 batches (b1, b2), 4 condition types and 2 condition labels (4x2 combinations of condition labels, jointly denoted as c1111, c1112, c1121, c1122, c1211, c1212, c1221, c1222, c2111, c2112, c2121, c2122, c2211, c2212, c2221, c2222)

We held out the data under condition c22 (2 condition types)/condition c222 (3 condition types)/condition c2222 (4 condition types), and trained the model on the remaining data. We then use the training data under condition c11 (2 condition types)/c111 (3 condition types)/c1111 (4 condition types) and batch b1 to predict the data under the held-out condition (c22/c222/c2222) and batch b1. We measure the perturbation prediction accuracy using MSE and R2 scores. We then measure the condition-associated key genes (CKGs) detection accuracy using AUPRC. The final score:

Figure R18. The perturbation prediction accuracy (measured with MSE, R2) and the CKGs detection accuracy (AUPRC) of the model with different condition types.

The final score shows that the model performance decreases with the increase of condition types. More condition types make it harder for scDisInFact to learn the perturbation effect of individual condition types, which results in the deterioration of the overall model performance. Although the overall performance of the method still remains high under the 4 cond-types setting (R2 score above 0.88), we recommend no more than 4 condition types.

Q3.2: How many samples need to be collected for each batch/condition and what proportion of the condition/batch can be missing and rescued by the perturbation analysis?

A3.2: Missing count matrices of different conditions and batches affect the performance of the model (mainly perturbation prediction) in different ways. In order for scDisInFact to make accurate perturbation prediction, we have two rules on the dataset:

- a. The training count matrices must cover the conditions (not the condition combinations) and batches of the input and predicted count matrices. For example, when we predict the count under condition (stim, healthy, batch 0), the training data does not have to include the count matrix under (stim, healthy, batch 0), but should include at least one data matrix that has the stim condition, at least one data matrix that has the healthy condition at least one data matrix under batch 0.
- b. Enough count matrices should be provided to make sure that the effect of every condition (of different condition types) and batch can be separately learned from the data. For example, when the training data only includes (stim, healthy) and (ctrl, severe) conditions, the dataset only has the information of the joint effect of both conditions, but no information about the separate effect of each condition. Being trained on such data, the model cannot tell whether the difference of data distribution is contributed by stim/ctrl condition or healthy/severe condition. Similarly, when the training data only include (stim, batch 0) and (ctrl, batch 1), the model also cannot tell whether the difference of data distribution is contributed by stim/ctrl condition or the batch effect. To explain the requirement in a more precise format: For every two condition labels a and b of a certain condition type m , there must exist two count matrices in the dataset with all remaining condition types and batch the same but only the condition type m different (one matrix correspond to a , another matrix correspond to b).

In the manuscript, we tested how the model performance changed with the increase of missing matrices (Sec. 2.2.4, Pages 8-9, Fig. S6d) using datasets with 2 batches, 2 condition types and

2 condition labels for each condition type. In the test, we observe that with the increase of the number of missing matrices, the model performance deteriorates. In addition, there is a significant drop in model performance when the total number of missing matrices increases to 4. This is because when the missing matrix is 4, the *path* that connects different data matrices in the training data breaks (see Rule b above), where the condition effect of *healthy/severe* (Fig. S6c) is not differentiable from the batch effect. We have added the above discussion on the requirements of input matrices to the manuscript (main manuscript Lines 255-256, Supplementary Note 3).

Q3.3: And how many cells are needed for each condition/batch?

A3.3: This is a great question considering that deep learning models need a certain number of samples for the proper training and predictive power of the model. Generally speaking, more cells for each input count matrix is beneficial. Here we provide robustness analysis on how the number of cells per sample (i.e. per data matrix) affects the overall performance of the model.

We evaluate the model performance by measuring the perturbation prediction accuracy. We test our model on the same simulated datasets used in the manuscript (a total of 9 datasets), and subsample the total number of cells in the datasets, such that the resulting dataset has respectively 1000, 2000, 5000, and 10000 cells in total (corresponding to 125, 250, 625, 1250 for each count matrix). The perturbation prediction accuracy in terms of MSE on different sizes of datasets is shown as following:

Figure R20. Perturbation prediction accuracy of the model (measured with MSE) with different sample sizes.

We observe that with the increase of the total number of cells, scDisInFact shows stronger predictive power, with significantly lower MSE. Also, the performance has a clear drop from 625 cells per data matrix to 250 cells per data matrix. While determining an exact minimum cell count is challenging, we suggest aiming for a minimum of 500 cells per data matrix, as indicated by these findings.

4. If possible, I recommend the authors provide some theoretical discussions about the necessary assumptions to ensure the identifiability of the problems. This may help users to understand what assumptions were made when using this method.

We thank the reviewer for this suggestion. The major assumptions for perturbation prediction are summarized in our response to the reviewer's comment #1 (Pages 15-16 of this response file): (1) the batch effect is consistent across conditions, and (2) the batch/condition effects are additive in the embedding space. We discussed why these assumptions were made in the response, and we also incorporated the discussions on the assumptions made in this model in the Discussion section of the main manuscript (Page 15, Lines 431-448).

Reviewer #3

This work proposes a Variational Autoencoder-based method that aims to disentangle the batch, condition, and common effects of multi-batch and multi-condition single-cell RNAseq data in separate latent representations. The method was evaluated on simulated and real datasets to perform tasks including batch effect removal, condition-associated key gene detection, and perturbation prediction. Results indicate its superior performance to baseline methods including scINSIGHAT and scGen/scPreGAN.

In general, this disentanglement task is challenging, especially when batch and condition are one-on-one aligned such that none of the conditions are shared in more than one batch. Nonetheless, the approach proposed in this study sheds light on addressing the problem when conditions are shared across multiple batches.

I have a generally positive view of this work, yet there are still some concerns regarding the technical details.

1. The deep learning framework is trained with multi-tasking and in each iteration there are 3 consecutive steps for parameter update, which makes the training procedure complicated and challenging. Therefore, a detailed report showing the learning behavior of the model would be a helpful reference for replication or applying the method to new datasets. Key items include convergence curves of overall and separate losses, learning rate configuration, training and inference efficiency, etc.

We thank the reviewers for the suggestion. First, it is worth noting that we have updated our scDisInFact model according to the reviewers suggestions (Q7 of reviewer #1, Q1&2 of reviewer #2) and the ablation test results. We removed the redundant components of the model while keeping the original model performance. The updates on the model are summarized below (details of the updated model are included in the manuscript Pages 16-17 Lines 468-505):

1. We removed the discriminator and the total correlation loss.
2. We removed the contrastive loss.
3. We changed the prior distribution of unshared-bio factors from standard normal distributions into Gaussian Mixture distributions (each condition corresponds to one Gaussian, the means and variances of Gaussians are learned from the data).

The removal of the discriminator and total correlation loss means that the model does not need to be trained in an adversarial manner. We removed the third step that updates the discriminator, which left the model with the first two consecutive steps.

Convergence of losses: here we post the learning curve of the model on the simulated and real datasets tested in our manuscript (Sec. 4.2, Page 18 Lines 521-523, Fig. S16, Supplementary Note 5). The learning curves show that scDisInFact converges on all training datasets at the end of the training stages.

Figure R21. The learning curves of scDisInFact on simulated and real datasets.

Learning rate configuration: In the training procedures of all models above, we used the adam optimizer, with learning rate $5e-4$, and batch size 64.

Training efficiency: We tested the training efficiency by measuring the running time of the model on datasets of different sizes. The running time test result is shown in Figure R12, where scDisInFact scales well with the increase of data size.

2. The multi-tasking loss weights vary a lot, some are very small (e.g., $1e-5$, $1e-4$) compared to others (1.0). A sensitivity analysis of the hyper-parameters and/or an ablation study of some of the auxiliary losses might be helpful to understand the importance of different losses.

This is a good point. We conducted the ablation study and the hyper-parameter tests according to the requirement of the reviewer:

1. Ablation test

This comment is related to some comments from other reviewers (Q7 of reviewer #1, Q1&2 of reviewer #2, Q2 of reviewer #3), and we address these comments together.

After conducting careful ablation tests on the loss terms of our original model, we learned that there is redundancy in our original model. Therefore, we removed the redundant parts of the model while maintaining the performance of the model. The updates on the model are summarized below:

1. We removed the discriminator and the total correlation loss. Our ablation test does not support that the total correlation loss improves the overall performance of the model, and the independence assumption between shared-bio factors and unshared-bio factors enforced by the total correlation loss is too strong.
2. We remove the contrastive loss. Given the existing classifier in the model, the additional contrastive loss does not further improve the overall performance, especially the perturbation prediction accuracy. The ablation test did not show the advantage of including the contrastive loss.
3. We changed the prior distribution of unshared-bio factors from standard normal distributions into Gaussian Mixture distributions (each condition label corresponds to one Gaussian, and the means and variances of Gaussians are learned from data). The standard normal prior distribution does not match the true prior of the unshared-bio factor, where factors of different conditions should be distinguished.

The details of the updated model are included in the manuscript (Pages 16-17 Lines 468-505). After updating the model, we re-ran all the tests in the manuscript, and replaced the results and the corresponding figures. The overall performance of the model remains the same, but the complexity of the model is reduced.

We conducted the ablation test on the updated model to justify the necessity of the remaining loss function. There are totally 5 loss terms in the updated model: the reconstruction loss, the KL divergence to the prior distribution, the MMD loss on the latent factors, the classification loss on the unshared-bio factor, and the group-lasso loss on the unshared encoders (Eq. 6 in the manuscript).

Since the reconstruction loss and the KL divergence loss (which together make the ELBO loss) are essential terms for a VAE model, we conducted the ablation test on the remaining three losses: MMD loss, classification loss, and group-lasso loss. We conducted the ablation test using the same simulated datasets used in the manuscript (Sec. 2.2, totally 9 datasets). In the model training procedure, we held out the counts under the condition $\langle ctrl, severe \rangle$, of batches 1&2, and trained the model using the remaining count matrices. We describe the ablation tests of each loss term below:

1. *MMD loss*. In scDisInFact, the use of MMD loss serves three purposes:

- a. Removing batch effects from shared-bio and unshared-bio factors, which means that the shared-bio/unshared-bio factors of different batches should follow the same distribution.
- b. Removing condition effects from shared-bio factors, which means that the shared-bio factors of different conditions should follow the same distribution.
- c. Disentangling unshared-bio factors of different condition types. That is to make sure that the unshared-bio factors of a certain condition type do not encode the condition effect of other condition types.

We use ASW scores (described in Sec. 4.9 in the manuscript) to measure the effectiveness of the MMD loss in terms of the three purposes above. Fig. R22a-c shows the ASW scores to measure respectively: (a) the alignment of shared-bio and unshared-bio factors across different batches (Fig. R22a); (b) the alignment of shared-bio factors across different conditions (Fig. R22b). (c) the alignment of unshared-bio factors across different conditions of other condition types (except for the condition type it encodes, Fig. R22c). These results show overall better ASW scores when using MMD loss (reg:0.0001) than not using MMD loss (reg:0), which means that the use of MMD loss helps achieve the three purposes above.

Figure R22. Ablation test of MMD loss (no MMD, reg: 0; MMD, reg: 0.0001) **a.** Alignment of shared-bio and unshared-bio factors across batches. **b.** Alignment of shared-bio factors across different conditions. **c.** alignment of unshared-bio factors across different conditions.

The use of MMD loss can further affect the accuracy of perturbation prediction. We test the accuracy of perturbation prediction with and without using MMD loss. We use count matrix of different conditions as input and predict the corresponding count matrix in the held-out condition (ctrl & severe & batch 0):

- a. Input stim & severe & batch 0, predict “Treatment”;
- b. Input ctrl & healthy & batch 0, predict “Severity”;
- c. Input stim & healthy & batch 0, predict “Treatment & Severity”;

Since we have the ground truth count matrix, we directly measure the MSE between the predicted counts and the ground truth counts for each cell, and report the average MSE. The result shows that the use of MMD loss improves the perturbation prediction accuracy (Fig. R23).

Figure R23. Perturbation prediction accuracy with MMD loss (reg: 0.0001) and without MMD loss (reg: 0).

2. The classification loss. The classification loss separates unshared-bio factors corresponding to different conditions, which is critical for the perturbation prediction accuracy of the model. We trained the model with and without classification loss (regularization weights 0 and 1), and calculated the perturbation prediction accuracy using MSE. The results show that the model with classification loss (reg: 1) shows lower MSE compared to the model without classification loss (Fig. R24).

Figure R24. Perturbation prediction accuracy (measured with MSE) of the model with and without classification loss (reg: 0 & reg: 1)

3. *The group-lasso loss.* The group-lasso loss helps the model to extract the most important CKGs. We trained the model with and without group-lasso loss, and calculated the CKGs detection using AUPRC (the same metric in the manuscript). As expected, the use of group-lasso loss (reg: 1) significantly improves the CKGs detection accuracy of the model (Fig. R25).

Figure R25. Perturbation prediction accuracy (measured with MSE) of the model with and without group-lasso loss (reg: 0 & reg: 1)

We have included the ablation test results (Figs. S12,S13) and corresponding discussions in the manuscript (Sec. 2.5.1, Supplementary Note 3).

2. Hyper-parameter test

We further conduct hyper-parameter tests to evaluate the robustness of the model performance under different hyper-parameter settings. The main hyper-parameters of the model include:

- 1) The regularization weight of MMD loss
- 2) The regularization weight of group-lasso loss
- 3) The regularization weight of classification loss
- 4) The regularization weight of KL divergence
- 5) The latent dimensions

We conducted the hyper-parameter test using the same simulated datasets used in the manuscript (totally 9 datasets). In the model training procedure, we held out the counts under the condition (ctrl, severe, batches 1&2), and trained the model using the remaining count matrices, similar to the ablation test. We evaluate the model performance in perturbation prediction and CKGs detection (Supplementary Note 4).

- 1) We test the model performance with different weights of MMD loss. We evaluate the model performance using perturbation prediction accuracy (Fig. R26) and CKGs detection accuracy (Fig. R27):

Figure R26. The perturbation prediction accuracy of *scDisInFact* with different MMD regularization weights. The accuracy is measured with MSE.

Figure R27. The CKGs detection accuracy of *scDisInFact* with different MMD regularization weights. The accuracy is measured with AUPRC.

The test result shows that the perturbation prediction accuracy increases with the increase of MMD loss, whereas the CKGs detection accuracy decreases with MMD loss. “Reg: 0.0001” shows a more balanced result.

2) We test the model performance with different weights of group-lasso loss. Again, we evaluate the perturbation prediction accuracy and CKGs detection accuracy (Figs. R28,R29).

Figure R28. The perturbation prediction accuracy of *scDisInFact* with different weights of group-lasso loss. The accuracy is measured with MSE.

Figure R29. The CKGs detection accuracy of *scDisInFact* with different weights of group-lasso loss. The accuracy is measured with AUPRC.

The test result shows that the CKGs detection accuracy of scDisInFact increases with the increase of group-lasso weight, whereas the perturbation prediction accuracy decreases with the increase of group-lasso weight. “reg: 1” shows a more balanced result.

3) We test the model performance with different weights of classification loss. Again, we evaluate the perturbation prediction accuracy and CKGs detection accuracy (Figs. R30,R31).

Figure R30. The perturbation prediction accuracy of scDisInFact with different weights of classification loss. The accuracy is measured with MSE.

Figure R31. The CKGs detection accuracy of scDisInFact with different weights of classification loss. The accuracy is measured with AUPRC.

The model with classification weight around “reg:0.1” to “reg:1” shows the best performance in both CKGs detection and perturbation prediction.

4) We then test how KL divergence weight affects the overall performance of the model. We mainly check the KL divergence weight of the unshared factors, which significantly affect the clustering of the latent space. The KL divergence weight of shared factors is set to be a very small value (1e-5) to encourage the separation of cell types of the shared embedding. We evaluate the perturbation prediction accuracy and CKGs detection accuracy (Figs. R32,R33).

Figure R32. The perturbation prediction accuracy of scDisInFact with different weights of KL divergence (unshared factors). The accuracy is measured with MSE.

Figure R33. The CKGs detection accuracy of scDisInFact with different weights of KL divergence (unshared factors). The accuracy is measured with AUPRC.

The model with weight from “reg: 0.0001” to “reg:0.01” shows the best performance in perturbation prediction and CKGs detection accuracies.

5) We test how the number of latent dimensions affects the overall performance of the model. We run the model with different numbers of shared and unshared latent dimensions. We evaluate the perturbation prediction accuracy and CKGs detection accuracy (Figs. R34,R35).

Figure R34. The perturbation prediction accuracy of scDisInFact with different latent dimensions. The accuracy is measured with MSE.

Figure R35. The CKGs detection accuracy of scDisInFact with different latent dimensions. The accuracy is measured with AUPRC.

Figs. R34-R35 shows that the model performance is not affected much by the latent dimensions. In all our tests in the manuscript, we used "shared: 8, unshared: 2" to run scDisInFact.

From the above results on all model hyper-parameters, we observe that in most of the cases, there is no drastic change in the performance of the model when changing the hyperparameters by magnitude, showing the robustness of the model against hyperparameter changes.

According to the result of the hyper-parameter test, we select the set of hyper-parameters ($\text{reg}_{kl} = 10^{-2}$, $\text{reg}_{\text{mmd}}=10^{-4}$, $\text{reg}_{\text{ce}}=1$, $\text{reg}_{\text{gl}}=1$, $\text{shared}=8$, $\text{unshared}=2$) as recommended hyper-parameters. We have included the above hyper-parameter test and recommended settings in the manuscript (Sec. 2.5.2, Pages 13-14, Lines 379-400, Supplementary Note 4).

3. The authors mentioned in Section 4.5 that “Users can manually tune the hyper-parameters around the recommended setting instead of conducting a comprehensive grid search, and it should provide a reasonable result on most of the dataset.”. Following the above comments, would the authors give more detailed instructions on how to diagnose model training and judge whether on a new dataset the model has succeeded or failed?

We thank the reviewer for the suggestion. In our updated manuscript, we have included more details on how to tune the hyper-parameters and how to diagnose if the model is trained successfully with the hyper-parameter setting (Sec. 4.5, Pages 19-20, Lines 559-574).

4. In 2.2.2 when comparing with scINSIGHT, training two models might result in unaligned latent representations, a fairer way might be to create a single new condition using Cartesian product of the original two conditions.

We thank the reviewer for the suggestion. In Sec. 2.2.2, we re-ran scINSIGHT using the Cartesian product of the original two conditions instead (including 4 combined conditions: “ctrl_healthy”, “stim_healthy”, “ctrl_severe”, “stim_severe”; description in Page 6, Lines 145-147). We updated the corresponding test results (Fig. 2d).

Other minor comments:

In Fig. 2d the middle sub-figure, the performance of scDisInfect looks worse than scINSIGHT with batch correction score, would the authors elaborate more on this result?

While scINSIGHT has a higher batch correction score than scDisInFact, both methods show extremely high scores (between 0.95 and 0.98), indicating that both methods learned latent embedding that aligns very well across batches and conditions. It is also critical to maintain cell type identities of cells during integration (measured by ARI), where scDisInFact clearly shows better performance than scINSIGHT.

The text in Section 4.5 describing model architecture might be better visualized in a compact diagram.

We thank the reviewer for this suggestion. We now have added the compact diagram in the main manuscript (Table S9).

References

- Lotfollahi, Mohammad, Anna Klimovskaia Susmelj, Carlo De Donno, Leon Hetzel, Yuge Ji, Ignacio L. Ibarra, Sanjay R. Srivatsan, Mohsen Naghipourfar, Riza M. Daza, Beth Martin, Jay Shendure, Jose L. McFaline-Figueroa, Pierre Boyeau, F. Alexander Wolf, Nafissa Yakubova, Stephan Günemann, Cole Trapnell, David Lopez-Paz, and Fabian J. Theis. 2023. "Predicting Cellular Responses to Complex Perturbations in High-Throughput Screens." *Molecular Systems Biology*, May, e11517. doi:10.15252/msb.202211517.
- Lotfollahi, Mohammad, F. Alexander Wolf, and Fabian J. Theis. 2019. "scGen Predicts Single-Cell Perturbation Responses." *Nature Methods* 16 (8): 715–21. doi:10.1038/s41592-019-0494-8.
- Pratapa, Aditya, Amogh P. Jalihal, Jeffrey N. Law, Aditya Bharadwaj, and T. M. Murali. 2020. "Benchmarking Algorithms for Gene Regulatory Network Inference from Single-Cell Transcriptomic Data." *Nature Methods*, January. doi:10.1038/s41592-019-0690-6.
- Tian, Yuan, Lindsay N. Carpp, Helen E. R. Miller, Michael Zager, Evan W. Newell, and Raphael Gottardo. 2022. "Single-Cell Immunology of SARS-CoV-2 Infection." *Nature Biotechnology* 40 (1): 30–41. doi:10.1038/s41587-021-01131-y.
- Wei, Xiajie, Jiayi Dong, and Fei Wang. 2022. "scPreGAN, a Deep Generative Model for Predicting the Response of Single Cell Expression to Perturbation." *Bioinformatics*, May. doi:10.1093/bioinformatics/btac357.

REVIEWERS' COMMENTS

Reviewer #1 (Remarks to the Author):

I appreciate that the authors addressed most of my comments. The manuscript has been improved. I just have a few remaining comments.

- In aligning partially matching cell types, two particular cell types (10 & 16) were removed. Is there any particular reason for this selection? Do cell type sizes (# of cells) affect partial alignment?
- The rare cell types are often unknown and likely specific to particular condition(s). So, it might be helpful to discuss how unaligned cells from the sophisticated scDisInFact method relate to these rare types. Ideally, the authors can compare differentially expressed genes (DEGs) of those cells to DEGs based on conditions to see if any interesting condition-related genes (or enrichments) show up.

Reviewer #2 (Remarks to the Author):

The authors did a very systematic work in addressing reviewers' questions. All my concerns have been eliminated data. Thanks.

Reviewer #3 (Remarks to the Author):

The authors have addressed all my comments by providing additional experimental results and explanations, I would hence suggest acceptance of the manuscript.

Responses to Reviewers' Comments

We thank the reviewers for their valuable time to review our revised manuscript and their supportive comments.

Please find below the reviewers' comments (in black) and our pointwise responses (in blue). The changes in the manuscript or supplementary material are marked in red. Figures in this response file are numbered Figure R1, Figure R2, etc.

Reviewer #1

I appreciate that the authors addressed most of my comments. The manuscript has been improved. I just have a few remaining comments.

1. In aligning partially matching cell types, two particular cell types (10 & 16) were removed. Is there any particular reason for this selection?

There is no particular reason for the selection of cell types 10 and 16. These two cell types were randomly picked for the partial alignment test. We included the details of these tests and how cell types were chosen to be removed in Supplementary Note 1. Here we show an additional test where we removed different cell types: cell type 3 (for 1 CT rem) or cell types 3 & 4 (for 2 CT rem). We obtained similar results (Figs. R1&R2, shown below) as those obtained by removing cell types 10 and 16 (Supplementary Fig. 3).

Figure R1. The ARI scores and ASW-batch scores of *scDisInFact* when (1) all cell types are matched across batches (All CT), (2) one cell type is removed from one batch (1 CT rem: remove cell type 3), and (3) two cell types are removed from one batch (2 CT rem: remove cell types 3 and 4).

Figure R2. The ARI scores and ASW-batch scores of scDisInFact when (1) all cell types are matched across conditions (All CT), (2) one cell type is removed from healthy condition (1 CT rem), and (3) two cell types are removed from healthy condition (2 CT rem).

To reflect this additional test, in Supplementary Note 1, we added the following sentence:

“We performed tests to create the scenario of unequal cell type compositions by removing other cell types (removing one cell type: cell type 3; removing two cell types: cell types 3 and 4) and obtained similar results as those shown in Fig. S3 (results not shown).”

Do cell type sizes (# of cells) affect partial alignment?

This is a great question. Here we design an additional test on how the size of mismatched cell types affects the model's overall performance.

Since in our existing simulated data the clusters are of similar size, we generated an additional simulated dataset (2 batches, 1 condition type with 2 condition labels) with 5 clusters of different sizes (total cluster size across batches and conditions: 326, 466, 651, 1001, 3250). We then iteratively remove one of the 5 clusters from Batch 1 to create a partial alignment task and apply scDisInFact. The sizes of mismatches are different when we remove different clusters.

We calculated the alignment accuracy of shared-bio factors using NMI, ARI, and ASW-batch scores, where NMI and ARI scores measure the separation of cell types, and the ASW-batch score measures the alignment of batches.

Figure R3. The NMI, ARI, and ASW-batch scores of shared-bio factors on partially aligned datasets with different sizes of mismatches. The x-axis shows the total number of cells removed in Batch 1.

Figure R4. The UMAP visualization of the shared-bio factors when the largest cell type (cell type 1, totally 1624 cells) is missing in batch 1. scDisInFact successfully deals with the mismatch of large cluster size.

Overall, the results show that the overall performance of scDisInFact is robust to the size of mismatched clusters. scDisInFact successfully worked with the scenario where the largest cell type was removed, which accounts for more than 50% of all the cells in the batch (Fig. R3 & R4).

2. The rare cell types are often unknown and likely specific to particular condition(s). So, it might be helpful to discuss how unaligned cells from the sophisticated scDisInFact method relate to these rare types. Ideally, the authors can compare differentially expressed genes (DEGs) of those cells to DEGs based on conditions to see if any interesting condition-related genes (or enrichments) show up.

We thank the reviewer for this suggestion. Indeed, unaligned cells can correspond to rare cell types or new cell types in certain conditions. Such new cell types often are differentiated from an existing cell type as the response to specific conditions. Therefore, we designed a simulation procedure to create the scenario of such rare cell types developed under certain conditions, and test: (1) if they will be detected as unaligned cells by scDisInFact; and (2) if the DEGs between the unaligned cells and the remaining cells can identify the unaligned cells as a biologically meaningful cell type, as opposed to errors or noise of the integration.

The simulation procedure is as follows: (1) We simulate datasets with 2 batches and 2 conditions ("ctrl" and "stim"), following the procedure described in the manuscript (Section 4.6, Pages 20-21); (2) We randomly picked one cell type (Cell Type 1 in condition "stim"), and selected a small sub-population of the cells within Cell Type 1 to create a rare cell type in

condition “stim”; (3) Cells in the rare cell type are “more perturbed” due to the condition effect than other cells in the original Cell Type 1 which make them a different cell type. We then applied stronger perturbation on the perturbed genes associated with the condition “stim” to these cells.

We applied scDisInFact to this simulated dataset. The visualization of the shared-bio factors is shown in Fig. R5. Then we ran the Leiden clustering algorithm on the shared-bio factors, and obtained rare cell types that only existed in “stim” condition (cluster 16 in the lower 4 plots in Fig. R5). This shows that scDisInFact identifies the rare cell type as unaligned cells, instead of aligning it with any existing cell type.

We then conducted DE analysis on the identified rare cell type, compared against remaining cells of the same condition and batch, using the Wilcoxon Rank Sum test. Then we compared the detected DE genes with the ground truth perturbed genes, and measured the detection accuracy using AUPRC and AUROC. The results are shown in the barplots in Fig. R6, where we can see that the detected DE genes highly overlap with the condition-related genes.

We have added this potential application of scDisInFact in detecting condition-specific rare cell types in the manuscript (Lines 178-180):

“Given the capability of scDisInFact in integrating datasets with mismatched cell type compositions and detection of rare cell types, it can potentially be used to reveal condition-specific rare cell types which appear to be unaligned cells after integration.”

Ground Truth Annotations

Leiden Cluster Result

Figure R5. The UMAP visualization of the shared-bio factor, where the cells are colored by ground truth cell type annotation and the Leiden clustering result. The rare cell type only exists in the “stim” condition and is within the red dashed circle.

Figure R6. The DE gene detection accuracy. The accuracy is measured with AUPRC and AUROC scores.

Reviewer #2

The authors did a very systematic work in addressing reviewers' questions. All my concerns have been eliminated. Thanks.

Reviewer #3

The authors have addressed all my comments by providing additional experimental results and explanations, I would hence suggest acceptance of the manuscript.